



# Testing Hypotheses About Glacial Dynamics and the Stage 11 Paradox Using a Statistical Model of Paleo-Climate

Robert K. Kaufmann[1], Felix Pretis[2]

[1]Department of Earth and Environment, Boston University, Boston, Massachusetts, USA, 02215
[2]Department of Economics, University of Victoria, Victoria, BC, Canada; and Nuffield College, University of Oxford, Oxford, UK

*Correspondence to*: Robert K. Kaufmann (Kaufmann@bu.edu)

**Abstract** To test hypotheses about glacial dynamics, the Mid-Brunhes event, and the stage 11 paradox, we evaluate the ability of a statistical model to simulate climate during the previous ~800,000 years. Throughout this period, the model simulates the timing and magnitude of glacial cycles, including the saw-tooth pattern in which ice accumulates gradually and ablates rapidly, without nonlinearities or threshold effects. This suggests that nonlinearities and/or threshold effects do not play a critical role in glacial cycles. Furthermore, model accuracy throughout the previous ~800,000 years suggest that changes in glacial cycles associated with the Mid-Brunhes event, which occurs near the division between the out-of-sample period and the in-sample period, are not caused by changes in the dynamics of the climate system. Conversely, poor model performance during MIS stage 11 and Termination V is consistent with arguments that the 'stage 11 paradox' represents a mismatch between orbital geometry and climate. Statistical orderings of simulation errors indicate that periods of reduced accuracy start with significant reductions in the model's ability to simulate carbon dioxide, non-sea-salt sodium, and non-sea-salt calcium. Their importance suggests that the stage 11 paradox is generated by changes in atmospheric and/or oceanic circulation that affect ocean ventilation of carbon dioxide.

## 1 Introduction

When considered over the last eight-hundred thousand years, climate shows highly persistent movements. Most notable are glacial cycles. During glaciations, temperature, greenhouse gas concentrations, and sea level remain below their sample mean for extended periods; during these same periods, land and sea ice remain above their sample means. These positions are reversed for extended periods known as inter-glacials. These persistent movements and complex climate dynamics create difficulties for statistical analyses of climate data over this long time-span. Using ordinary least squares to analyze time series that show persistent movements tends to indicate statistically meaningful relations among time series when none are present (Yule, 1929; Engle and Granger, 1987). Monte Carlo simulations indicate a relation (based on t statistics) for about 85 percent of random pairings of time series with highly persistent movements (Hendry and Juselius, 2000).

The difficulties posed by highly persistent movements and complex dynamics are greatly alleviated using the econometric methods of vector-autoregression, cointegration, and equilibrium correction. Using these methods, Kaufmann and Juselius (2013), herein KJ2013, estimate a statistical model of climate over the previous 391 thousand years. The model, termed a cointegration vector autoregression (CVAR), specifies four exogenous variables for orbital geometry; eccentricity, obliquity, precession, and summer time insolation at 65º south to simulate ten endogenous



variables that proxy various aspects of climate; Antarctic land and sea surface temperature, carbon dioxide and methane concentrations, land and sea ice, sea level, iron dust, and non sea-salt sulfate and calcium. The CVAR model explicitly represents long-run relations between climate and orbital geometry, which are given by ten cointegrating relations, and climate dynamics, which are given by the rates at which the climate system 'equilibrium corrects' from

disequilibrium in the long-run (cointegrating) relations. Davidson et al., (2016) apply a similar approach for a subset of climate variables.

These relations validate some basic hypotheses about the mechanisms that are postulated to drive glacial cycles (e.g. carbon dioxide affects temperature via radiative forcing), reproduce the main features of glacial cycles (e.g. the timing, magnitude, and saw-tooth pattern of changes in land ice volume), and separate observed deglaciations from skipped

obliquity/precession beats (e.g. Huybers, 2012), which are peaks in insolation, including obliquity that do not generate deglaciations (Huybers and Wunsch, 2005; Tzedakis et al., 2017). Subsequent analyses of the statistical model suggest a weak form of the Milankovitch hypothesis in which orbital geometry drives glacial cycles, with small perturbations imposed by internal climate dynamics (Kaufmann and Juselius, 2016).

Conclusions that are based on a model conditioned solely on orbital geometry are notable because many climate

models cannot simulate atmospheric concentrations of $CO_2$ (Archer et al., 2000). This has lead to hypotheses that orbital geometry and GHG are the 'two primary forcings' to the climate system (e.g. Yin and Berger, 2012). But KJ 2013 test and reject the hypothesis that carbon dioxide or methane is exogenous to the climate system; their concentrations are endogenous, driven by orbital geometry, which is exogenous to and is the primary driver of climate. Models that do simulate $CO_2$ endogenously cannot simulate other aspects of climate jointly (e.g. ice volume) and so

are simulated in absence of feedbacks (Brovkin et al., 2012) or in two steps (e.g. Ganopolski et al., 2016), which may cause models to understate the effects of changes in orbital position (Pretis and Kaufmann, in review).

Despite the strengths of the CVAR model, the resultant conclusions about the drivers of glacial cycles are tempered by the fact they are based on in-sample simulations over the previous 391 thousand years (i.e. the model simply reproduces the data from which it is estimated). A more rigorous methodology would use the four variables for orbital

geometry to simulate the ten climate/physical variables for the entire period for which proxy data are available, which spans the previous ~800 thousand years.

Here, we simulate the model reported by KJ2013 for the previous 800 thousand years, which corresponds to the entire period recorded by the Dome C core. We evaluate model performance by computing the root mean square of the simulation errors (RMSE) and identifying periods when differences between simulated and observed values are

statistically significant. These measures are used to test three hypotheses about glacial dynamics that have been discussed in the literature (see section 4 for discussion):

1.  Nonlinearities, threshold effects, or phase-specific governing equations play an important role in the timing and magnitude of glacial cycles.

2.  The Mid-Brunhes event (MBE), which refers to a climatic shift that occurs during the transition between

marine isotope stage (MIS) 12 and MIS 11 (Jansen et al., 1986), changes the dynamics that drive glacial cycles.

3.  The 'stage 11 paradox' represents a mismatch between orbital geometry and climate.



Values for the RMSE and statistical differences between simulated values and values from the proxy record indicate that the model generally performs well during the in- and out-of-sample period. We interpret this general accuracy to indicate that:

1.    Nonlinear relations,threshold effects, and/phase-specific governing equations do not play a critical role in
glacial cycles.

2.     Glacial cycles are driven by the same dynamics before and after the MBE.

3.    Terminations in general - and the 'stage 11 paradox' in particular - may be caused by changes in atmospheric circulation and/or the extent of sea ice, which affects the ventilation of the deep ocean and ultimately, affects the atmospheric concentration of carbon dioxide.

These results and the methods used to obtain them are described in five sections. Section 2 describes the data and methods used to generate and analyze the simulations. The results are described in section 3. Section 4 interprets the results relative to the three hypotheses described previously, and section 5 concludes.

## 2 Methods

The CVAR model described by KJ2013 is simulated in a dynamic simulation (equivalent to a dynamic forecast) conditioned on orbital geometry alone over the 791 thousand years before the present (kyr BP). Simulated values ($\hat{x}_t$) are subtracted from the corresponding values from the proxy record ($x_t$) to calculate simulation errors $\varepsilon_t = x_t - \hat{x}_t$. Simulation errors ($\varepsilon_t$) are analyzed three ways. First, we compute the root mean square error (RMSE) to evaluate model accuracy over pre-defined periods. Second, simulation errors are analyzed to identify periods when the model
fails systematically, either in a single time step (outlier) or during two or more consecutive time steps (persisting errors). Third we examine the statistical ordering among simulation errors (and the explanatory power of simulations that are generated by conditioning the model on endogenous variables) to evaluate competing hypotheses for the 'stage 11 paradox,' which is a significant mismatch between orbital geometry and climate associated with marine isotope stage (MIS) 11, 424 – 375 kyr BP (Imbrie et al., 1993).

### 2.1Model Data

The four series used to represent orbital position, the six series used to represent climate, and the four series used to represent physical and biological mechanisms that link the six climate variables to each other and orbital geometry
are the same as those used in KJ2013 (Table 1). KJ2013 uses four series to represent the effect of orbital geometry: precession (*Prec*), obliquity (*Obl*), eccentricity (*Ecc*), and summer-time insolation at 65ºS (*SunSum*). Observations for these time series are compiled back to 800 kyr BP from the same sources used by KJ 2013 (Paillard, 1996).
KJ2013 uses these four measures of orbital geometry alone to simulate ten endogenous variables (six climate and four mechanisms); climate variables include land surface temperature (*Temp*), the atmospheric concentration of carbon
dioxide ($CO_2$) methane ($CH_4$), sea surface temperature (*SST*), land ice volume (*Ice*), and sea level (*Level*). Variables that capture mechanisms include iron dust (*Fe*), sea-salt sodium (*Na*), non sea-salt sulfate ($SO_4$), and non sea-salt calcium (*Ca*); for additional details about the each series see Section I of the Supplemental Material.



Data for *Temp*, *CO₂*, and *CH₄* are obtained from cores drilled into the Antarctic ice sheet. Carbon dioxide and methane are well-mixed gases and so measurements from Antarctic ice proxy global concentrations. *Temp* represents local conditions, but can be converted to global values by assuming that a scaling factor, which is derived from a limited set of observations can be applied across all observations (Masson-Delmotte et al., 2010; Masson-Delmotte et al., 2006). The $\delta^{18}O$ data that are used to proxy ice volume, which also includes information about deep water temperature (Chappell and Shackleton., 1986; Shackleton, 2000), are derived from 57 cores drilled by the Deep-Sea Drilling Project and Ocean Drilling Program across the globe (Lisiecki and Raymo, 2005). Sea surface temperature is constructed using alkenones from site PS2489-2/ODP1090 in the sub-Antarctic Atlantic. Data for sea level are reconstructed using oxygen isotope records from Red Sea sediments (Siddall *et al.,* 2003).

These six variables are linked to each other and orbital position via physical and biological mechanisms that are represented by the four proxy variables. *Fe* is derived almost entirely from terrestrial sources and proxies changes in atmospheric circulation and a so-called iron fertilization effect, which may enhance the biotic uptake of $CO_2$ (Martin, 1990). Sulfate *SO₄* originates mainly from marine biogenic emissions of dimethylsulphide (after removing sea-salt sources using the Na data), and so proxies marine biological activity (Cosme et al., 2005). It is included to represent the possible effect of iron-containing dust on biological activity and/or the effect of biological activity on atmospheric $CO_2$. Sea salt sodium *Na* is derived from the sea-ice surface and proxies the extent of winter sea-ice (Wolff et al 2003). It is included to represent the possible effect of sea ice on the flow of $CO_2$ from the ocean to the atmosphere (Stephens and Keeling, 2000). Non sea-salt calcium *Ca* has a terrestrial origin (mainly Patagonia) and may represent changes in temperature, moisture, vegetation, wind strength, glacial coverage, or changes in sea level in and around Patagonia (Basile et al., 1997), a locale thought to play an important role in glacial cycles.

To make these data amenable to a statistical analysis, we convert them to a common time scale (EDC3) using conversions from Parrenin et al., (2007) and Ruddiman and Raymo (2003). Unevenly spaced observations are interpolated (linearly) to generate a data set in which each series has a time step of 1 kyr (Miller, 2019). To eliminate the effects on inverting matrices with elements that differ greatly in size (due to different units of measurement), each of the fourteen time series is standardized as follows:

$$x_i = (y_i - \bar{y})/\sqrt{Var(y)}, \qquad t = 1, \dots, 391 \tag{1}$$

where $y_t$ is the value (in original units), $\bar{y}$ is the average value over the in-sample period, and $Var(y)$ is the variance over the in-sample period.

## 2.2 Simulating the CVAR Model

The equations used to estimate the CVAR model in KJ2013 are given by:

$$\Delta x_t = A_0 \Delta w_t + A_1 \Delta w_{t-} + \Gamma_1 \Delta x_t + \Pi z'_{t-1} + \varepsilon_t \tag{2}$$

in which $x_t$ is a $10 \times 1$ vector that includes the ten endogenous variables; *Temp*, $CO_2$, $CH_4$, *Ice*, *Fe*, *Na*, *Ca*, $SO_4$, *Level*, and *SST*; $w_t$ is a $4 \times 1$ vector that includes the four exogenous variables *Ecc*, *Prec*, *Obliq*, and *SunsumS*; $z' = [x'_t, w'_t, 1]$, $\Gamma_1, A_0, A_1$ are $10 \times 14$ matrices of short-run coefficients; $\Pi$ is a $10 \times 15$ matrix of long-run coefficients,




$\Delta$ is the first difference operator ($\Delta x_t = x_t - x_{t-1}$), $\varepsilon_i$ is an error term with mean value zero and variance $\Omega$ that is normally, independently, and identicially distributed.

The condition that the conditional process ($x_t | w_t$) is nonstationary is formulated as a reduced rank hypothesis on the matrix $\Pi$

$$\Pi = \alpha\beta^{'} \tag{3}$$

in which $\alpha$ is a $10 \times r$ matrix of coefficients, which describe the rate at which the ten climate variables adjust back towards equilibrium after the system has been pushed away by exogenous shocks (i.e. changes in orbital geomtery); $r$ is the number of cointegration relations given by the reduced rank of the $\Pi$ matrix; and $\beta$ is a $r \times 15$ matrix of cointegration coefficients that define the $r$ stationary deviations from long-run equilibrium relationships, the so called

cointegration relations, $\beta'z_t$. Maximum likelihood estimates for the elements of the $\beta$ and $\alpha$ matrices as reported by KJ 2013 are given in section II of the Supplemental Material. The model in KJ2013 is estimated as a partial system (Johansen 1992, Harbo et al., 1998, Juselius 2006) where orbital variables are weakly exogenous.

Here we simulate the estimated model model over the full time period using a dynamic simulation in an open model, conditioned on the (strongly) exogenous orbital variables $w_t$ (equivalent to a dynamic forecast). To simulate climate

during the in- and out-of-sample periods, the ten endogenous variables $x$ are expressed as a function of the exogenous solar variables and shocks to the climate system by inverting Equation (2) into the moving average form:

$$x_t = C \sum_{i=1}^{t} \varepsilon_i + C^*(L)\varepsilon_t + C_w w_t + C_w^*(L)\Delta w_t \tag{4}$$

where $C = \beta_\perp (1 - \Gamma_1)^{-1} \alpha_\perp$; $\alpha_\perp$ is a $10 \times (10 - r)$ matrix orthogonal to $\alpha$ describing the stochastic trends and $\beta_\perp$ is a $10 \times (10 - r)$ matrix orthogonal to $\beta$ determining how the stochastic trends load into the climate variables; $L$ is

the lag operator (for example, $L\varepsilon_t = \varepsilon_{t-1}$); $C^*(L)$ and $C_w^*(L)$ are stationary lag polynomials; $C_w$ is $10 \times 4$; and the matrices are functions of the parameters ($A_0, A_1, \Gamma_1, \alpha, \beta$). Based on the ten cointergating relations reported by KJ2013 $r = 10$, then $C = 0$, the in- and out-of-sample simulations are based on model (2) subject to (3) by setting $\varepsilon_t = 0$ which implies that the simulated variables, $\hat{x}_t$, are calculated from the exogenous drivers, $C_w w_t$, ($A_0\Delta w_t$), the dynamics attached to them, $C_w^*(L)\Delta w_{t-1}$, ($A_1\Delta w_{t-1}$), and the internal climate dynamics $C^*(L)\varepsilon_t(\Gamma_1\Delta\hat{x}_{t-1}, \alpha\beta'z_{t-1})$.

The out-of-sample simulation is generated by allowing the model to 'spin up' between 800 kyr BP and 792 kyr BP, which enables the endogenous variables to converge towards the values that are implied by the exogenous conditioning variables (*Prec*, *Ecc*, *Obl*, and *SunSumS*). During this 'spin-up' period, the model is initialized using observed values for *Temp, SST*, and *Ice*, which are available starting 800 kyr BP. The time series of $CO_2$ $CH_4$, *Fe, Na, SO4, Ca*, and *Level* have more recent start dates (Table 1). For these variables, the model is initialized with values that correspond

to their sample mean. Once the model is spun-up, the model is run continuously through the present; values from 792 kyr BP through 392 kyr BP constitute the out-of-sample period. Values from 391 kyr BP through the present constitute the in-sample period.

**2.3 Statistical Measures of Model Performance**


We use RMSE as a simple heuristic to compare the model's predictive accuracy during the in- and out-of-sample periods. Because accuracy may vary over time, we use an indicator saturation technique [R-package *gets* Pretis et al., 2018; Castle et al., 2015] to identify periods during which the simulation significantly deviates from observations (i.e. simulation errors are statistically different from zero). Outliers refer to a statistically significant difference in the



simulated value of variable *x* relative to the observed value for a single time step, while persisting errors are statistically significant differences that persist for two or more consecutive time-steps. Outliers and persisting errors are evaluated for every possible time step. Here, we retain only those outliers or persisting errors that exceed the pα = 0.001 threshold. This tightly controls the false-positive rate of detected periods of model failure. The method used to identify outliers

and persisting errors are summarized in Supplementary Section III. This approach is used to assess the time-varying performance of climate models (Pretis et al., 2015), the forecast accuracy of economic predictions (Ericsson 2017), as well as to detect volcanic eruptions in temperature reconstructions in both simulated climate data (Pretis et al., 2016) and proxy-reconstructions (Schneider et al., 2017).

**2.4 Identifying Periods of Simulation Failures**

If model performance does not change over time, we expect outliers and persisting errors to occur randomly throughout the sample and be equally likely in each sub-sample. We use this assumption to compare the distribution of outliers and persisting errors between in-sample and out-of-sample periods and among nineteen marine isotope

stages. For each thousand-year time step, we count the number of variables that exhibit an outlier or persisting error. Following this procedure, the maximum number of outliers or persisting errors for any single time-step is ten. These sums (and values for individual variables) are assigned to the in- or out-of-sample period or individual marine isotope stages.

To evaluate the distribution of outliers and persisting errors between the in- and out-of-sample periods and among

marine isotope stages, we test whether their occurrence is different from a uniform random distribution (expected under the null-hypothesis of equal performance) using a Pearson chi-square test (P), which is calculated as follows:

$$P = \sum_{j=1}^{n} \frac{(O_j - E_j)^2}{E_j}$$    (5)

in which *n* is the number of periods (n=2; in-sample *j = 1*; out-of-sample *j = 2;* or nineteen marine isotope stages), $O_j$ is the number of outliers or persisting errors that are identified in period *j*, and $E_j$ is the number of occurrences expected

in period *j*.

The number of occurrences expected in period *j* ($E_j$) is calculated based on the null hypothesis that outliers or persisting errors are distributed uniformly among periods. This null implies that the expected value ($E_j$) can be calculated as:

$$E_j = \frac{Yr_j}{\sum_i^n Yr_{ji}} \times \sum_{j=1}^{n} O_j$$    (6)

in which *Yr* is the number of thousand-year time steps in period *j* for which observed values are available and *n* is the

number of periods for which observed values are available for the 791 kyr simulation period. *P* is evaluated against a $\chi^2$ distribution with n-1 degrees of freedom. If the test rejects the null hypothesis that outliers or persisting errors are distributed randomly among periods (i.e. some periods are simulated more/less accurately than others), the more accurate subsample is identified by the numerator of Equation (5) ($O_j - E_j$). A negative value during the in-sample period (($O_1 - E_1$) < 0) would indicate that the number of outliers or persisting errors detected during the in-sample

period is less than expected by a uniform random distribution. This result would suggest that the model generates a more accurate simulation during the in-sample period. Equations (5) and (6) also are used to test whether outliers or persisting errors are distributed randomly across the nineteen marine isotope stages (n=19) that fall within the 791 kyr simulation. The first observation is 791 kyr BP, which falls in MIS 19.



### 2.5 Causes for Model Failure

To evaluate the cause(s) for model failure, we test whether poor performance 'starts' with a specific variable(s) and whether this failure is communicated to the other variables through long- and short-run relations among endogenous variables. To identify the variable(s) that initiates the poor performance, we formalize techniques that are used by previous analyses. Previous analyses estimate a regression equation that specifies a dependent variable as a function of lagged values for an independent variable thought to 'precede' the dependent variable. For example, Li et al., (1998) conclude that $CO_2$ 'precedes' $\delta^{18}O$ based on regression results that indicate $\delta^{18}O$ is related to five lagged values of $CO_2$.

But this approach is incomplete (from a statistical perspective) because it ignores the autocorrelation structure of the dependent variable. To account for this effect, we use a technique developed by Granger (1969) that is used to analyze relations among climate variables during the instrumental temperature record (e.g. Kaufmann and Stern, 1997; Stern and Kaufmann, 2014). For this application, we estimate the following regression:

$$\varepsilon_{i,t} = \alpha + \sum_{i=1}^{10}\sum_{j=1}^{s} \phi_{i,j}\,\varepsilon_{i,t-j} + \sum_{i=1}^{10}\sum_{j=1}^{s} D_{i,t-j}\theta_{i,j}\,\varepsilon_{i,t-j} + \sum_{j=0}^{S}\pi_j\omega_{i,t-j} + \eta_{i,t} \tag{7}$$

in which $D_{i,t}$ is an indicator variable that equals one if the simulation error for variable $i$ during period $t$ is statistically different from zero (i.e. $\varepsilon_{i,t}$ is a persisting error) ($D_{i,t} = 0$ otherwise), $\eta$ is an error term (assumed to be normally distributed), and $\alpha, \phi, \theta, \pi$, are regression coefficients that are estimated using ordinary least squares. The number of lags (s) is determined using the Akaike Information criterion (Akaike, 1973). Equation (7) is estimated ten times, once with the simulation error for each endogenous variable on the left-hand side. We expect the coefficients $\phi_i$ generally to be statistically different from zero because simulation errors generally are correlated across variables, however, we are interested whether during the periods of simulation failure (as given by $D_{i,t} = 1$), persisting errors for other endogenous variables propagate through the system, pre-dating/predicting persisting errors in the endogenous climate variable being modelled. Because the level of significance of selection in the first stage ($p_\alpha \simeq 0.001$) makes false-positives in $D_{i,t}$ unlikely (approximately 1 outlier to be expected spuriously on average), the detection of breaks in the first stage probably has little effect on tests on $D_{i,t}$ in this second stage. We repeat this process without the simulation errors for sea level because the first observation for sea level (462 kyr BP) is much more recent than the other time series (Table 1), which limits the sample range when all ten simulation errors are analyzed using Equation (7).

For each simulation error for variable i ($\varepsilon_i$), we estimate Equation (7) ten times. In each, we eliminate the simulation errors for one of the ten endogenous variables interacted with its non-zero mean simulation dummy $\sum_{j=1}^{S} D_{i,t-j}\theta_{i,j}\varepsilon_{i,t-j}$. This restriction is evaluated using an F-statistic that tests the null hypothesis that the persisting errors for the endogenous variable eliminated from Equation (7) have no information about the dependent variable beyond the additional variables included. These variables include the lagged values of simulation errors, the persisting simulation errors for the other endogenous variables, and the four exogenous variables for orbital geometry. Rejecting this null hypothesis allows us to state that the model's inability to simulate the endogenous variable that is eliminated from Equation (7) (as indicated by persisting errors) precedes the simulation errors for the endogenous climate variable on the left-hand side of Equation (7).



## 3 Results

### 3.1 Model Performance

For both the in- and out-of-sample periods, Figure 1 suggests that the model generally captures the timing and magnitude of persistent changes in climate that are described by glacial cycles, which frequently are summarized by changes in land ice volume (*Ice*). For this variable, the model generally simulates the timing and magnitude of glaciations and terminations, including the gradual accumulation of ice and its rapid ablation (i.e. the saw-tooth

pattern). Furthermore, there are no skipped obliquity/precession beats (other than MIS 11). Finally, the model's ability to simulate glacial cycles during the out-of-sample period is inconsistent with speculation that the CVAR model's ability to reproduce the ten climate/physical variables during the in-sample period simply reflects the model's ability to reproduce the data used to estimate the coefficients. Instead, the ability of the model to simulate climate during the out-of-sample period suggests that its coefficients capture relations among orbital geometry and the ten

climate/physical proxies that govern the climate system beyond the sample period.

### 3.1.1 In- vs. Out-of-Sample Comparisons

The similarity between the model's accuracy in- and out-of-sample (Figure 1) is consistent with comparisons of root

mean square error (Figure 2). As expected, the RMSE for the out-of-sample period generally is larger than the RMSE for the in-sample period. But much of this increase is associated with MIS 11, most of which occurs during the out-of-sample period (Figure 1). If we eliminate MIS 11 from the out-of-sample period, the RMSE of the in- and out-of-sample periods are similar (Figure 2). The outsized effect on the RMSE for the out-of-sample period is consistent with the 'stage 11 paradox.'

Tests indicate that we cannot reject the null hypothesis that outliers are distributed randomly between the in- and out-of-sample periods (Table 2). A test statistic $\chi^2(1) = 0.09$ fails to reject (p > 0.76) the null hypothesis that as a group, outliers for the ten climate/physical variables are distributed randomly between the in- and out-of-sample periods. Conversely, a test statistic $\chi^2(1) = 52.5$ rejects (p < 0.001) the null hypothesis that as a group, persisting errors for the ten climate/physical variables are distributed randomly between the in- and out-of-sample periods.


### 3.1.2 Comparisons Among Marine Isotope Stages

Outliers and persisting errors are not distributed randomly among the nineteen marine isotope stages (Figure 3, Table 2). This result is generated in part by the 'stage 11 paradox.' If this stage is eliminated from consideration, we cannot

reject the null hypothesis that outliers for variables other than methane are distributed randomly among the remaining eighteen stages. Similarly, the RMSEs across variables are very similar in and out-of-sample when MIS 11 is excluded (Figure 2). Conversely, the number of persisting errors is not distributed randomly among the nineteen marine isotope stages, even if errors in MIS 11 are excluded (Table 2).



### 3.2 Causes for Model Failure

Applying the $p = 0.05$ threshold to the tests that evaluate restrictions on Equation (7), sixteen of the one hundred tests reject the null hypothesis that lagged values for persisting errors (interacted with the non-zero dummy variable D) have no information about current values for the simulation errors on the left-hand side of Equation (7) beyond the right-hand side variables that remain in Equation 7 (Table 3). For the eighty-one tests run on the nine endogenous variables (other than sea level), the null is rejected eleven times (Table 4). In both cases, the number of rejections observed is greater than the number expected due to repeated testing at $p = 0.05$, five and four rejections, respectively. Together, these results suggest that the test results reveal information about the statistical ordering of simulation errors.

### 4 Discussion

#### 4.1 Nonlinearties and/or threshold effects drive the timing and magnitude of glacial cycles

A recent review of terminations states "Terminations clearly represent a strongly nonlinear response to regional changes in the seasonality of solar radiation (Past interglacials Working Group of Pages, 2016)." We test this statement by using the CVAR to evaluate hypotheses about the importance of thresholds (e.g. Paillard, 1998; 2001; Ganopolski et al., 2016; Tzedakis, et al., 2017), nonlinearities (e.g. Tziperman et al., 2006), or governing equations that vary by phase of the glacial cycle. If any of these play an important role, the CVAR model, which does not include their effects, will not be able to simulate glacial cycles.

The CVAR model is largely linear. Both long- and short-run relations among variables are linear. The only non-linear relation is given by the fractional rate at which variables adjust to disequilibrium in the long-run relations ($\alpha$). But this nonlinearity is constrained by the fact that the fractional rate of adjustment is constant and applies during all phases of the glacial cycle.

Despite its largely linear specification, the CVAR generally simulates the timing and magnitude of changes in ice volume (and other variables) without any skipped beats other than stage 11. Furthermore, this linear specification allows the model to simulate the saw-tooth pattern by which ice volume builds slowly but melts rapidly. These results suggest that non-linear relations, thresholds, or changes in governing equations are not important drivers of glacial cycles. This suggestion does not reject their presence, rather, Occam's razor implies that nonlinearities, threshold effects, and/or phase-specific governing equations are not needed to simulate important aspects of glacial cycles.

Furthermore, the CVAR's ability to simulate climate during the out-of-sample period is inconsistent with the hypothesis that "glacial cycles would exist even in the absence of the insolation changes (Tziperman et al., 2006)." If glacial cycles exist independently of changes in orbital geometry, a statistical model that is conditioned only on orbital geometry and spun up with no memory of previous cycles would not be able to simulate glacial cycles accurately during the initial out-of-sample period. As in Gonapolski and Calov (2011), the ten variables come to an equilibrium and do not change thereafter if orbital geometry is held constant. Furthermore, the accuracy of the out-of-sample simulation is inconsistent with the argument that changes in solar insolation account for less than 20 percent of the variance in glacial temperature records (Wunsch, 2004).



### 4.2 The Mid Brunhes Event

The demarcation between the in- and out-of-sample period (391 kyr BP) falls close to the Mid-Brunhes event, MBE (Jansen et al., 1986). Compared to the in-sample period used to estimate KJ2013, the pre-MBE out-of-sample period
has; (1) lower concentrations of $CO_2$, (2) glacial cycles with a smaller amplitude, and (3) cooler but longer interglacial periods (EPICA, *et* al., 2004; Luthi et al., 2008; Hoenisch et al., 2009). These three changes beg the question, do they represent a change in the dynamics that drive glacial cycles and/or a change in the drivers of glacial cycles. The latter is supported by Yin (2013), who concludes, "through a set of internal mechanisms insolation alone induces a systematic difference between the interglacials before and after the 430 kyr ago in some ocean processes that are
critical for the carbon cycle." Conversely, Tzedakis et al., (2009) argue 'astronomical forcing alone cannot explain the difference in interglacial intensity before and after the MBE."
Our model simulations contradict the latter, that the MBE represents a change in the dynamics that drive glacial cycles. As indicated in Figure 1, the single set of relations among orbital geometry and the climate system embodied in the CVAR model simulates the different characteristics of glacial cycles before and after the MBE. As such, the MBE is
not a transition between regimes; rather there is something unique about the MBE in particular and MIS 11 in general.

### 4.3 Mechanisms for the Stage 11 Paradox

Imbrie et al., (1993) describe 'the stage 11 paradox' as a significant mismatch between orbital position and changes
in climate associated with MIS 11 in general and termination V (430 -415 kyr BP) in particular. The latter is defined by the maximum in benthic $\delta^{18}O$ of MIS 12 and the benthic $\delta^{18}O$ plateau of MIS 11 (Broecker and van Donk, 1970). These periods are unique: Termination V is the longest of any during the previous half million years (Berger and Loutre, 1996; Droxler et al., 2003; Loutre and Berger, 2003; McManus et al., 2003; EPICA Community Members, 2004; Rohling et al., 2010; Liseicki and Raymo, 2005; Ruddiman, 2007). MIS 11 also is the longest period of
prolonged, stable warm climate in the North Atlantic (Oppo et al., 1998; McManus et al*.,* 1999; 2003). Finally, many areas have air and sea surface temperatures that reach values consistent with interglacial periods even though large areas of the Earth's surface are covered by ice (Ruddiman, 2007). Despite these large changes in climate, the changes in orbital geometry are small.
Consistent with this seeming mismatch, the CVAR model does a poor job of simulating termination V in particular
and MIS 11 in general. Figures 1-3 indicate that MIS 11 has more variables with persisting errors than any other period, either in- or out-of- sample (as well as driving the higher RMSE out-of-sample). This indicates MIS 11 is a prolonged period during which the model is not able to use the four variables for orbital geometry to simulate climate, which is the definition of the 'stage 11 paradox.'

### 4.3.1 Difficulties in orbital tuning

The CVAR's model's poor performance during MIS 11 could be caused by difficulties in orbital tuning. The insolation peak for MIS 11 occurs in the middle of the warm stage therefore, orbital tuning delays the interglacial peak in $\delta^{18}O$ compared to other stages (Candy et al., 2014; Imbrie et al., 1984; Liseicki and Raymo, 2005). Furthermore, MIS 11



contains fewer tie points that can be used to anchor the chronology (Desprat et al., 2005), which means that the orbitally tuned chronology of MIS 11 is less secure than other warm stages (Candy et al., 2014). As such, the model's failure during this period may simply represent the poor quality of the chronology to which the simulation is compared. To evaluate whether the stage 11 paradox is an artifact of the poor quality of the chronology, we condition the model

on some of the endogenous variables that are thought to play an important role in glacial cycles. Conditioning a model on observed values for one or more endogenous variables always will improve performance (Oreskes et al., 1994), but the variable used to condition the model will have little effect on model performance if the model's poor performance during MIS 11 is caused by the poor quality of the chronology because no endogenous variable will have more/less information about the poor chronology. Contrary to this expectation, model performance during stage 11

depends on the variable used to condition the model. Conditioning the model on observed values of $CO_2$ or $Na$ allows the model to simulate more of the decline in $Ice$ (and more accurately simulate other variables) throughout MIS 11, including termination V (Figure 4). Conversely, conditioning the model on observed values for $SST$, which is thought to play an important role in MIS 11 (see below), does not improve the model's ability to simulate the interglacial in MIS 11. Although this failure may be explained by stronger latitudinal or meridional gradients in sea surface

temperature (Kandiano et al., 2012), large variations in accuracy that depend on the endogenous variable used to condition the model suggest that the model's failure during MIS 11 is not caused solely by weaknesses in orbital tuning.

### 4.3.2 Mechanistic Explanations

The mechanisms and sequences that generate the 'stage 11 paradox' cannot be fully identified by the CVAR model because it greatly simplifies physical relations and it has a relatively coarse temporal resolution (1 kyr). Conversely, its ability to accurately simulate glacial cycles (except MIS 11) using orbital position alone allows the CVAR model to test competing hypotheses about the 'stage 11 paradox' by identifying exceptions to the model sequences that

accurately simulate terminations other than termination V. In other words, the statistical ordering of simulation errors allows us to identify what is unique about MIS 11 (and termination V) and whether these differences play an important role.

Explanations for terminations in general - and stage 11 in particular - share several components. Many start with a change in meridonal overturning circulation and a bipolar seesaw that create a negative correlation between changes in hemispheric temperatures. Specifically, terminations may start with changes in orbital position that add freshwater

to the North Atlantic, this freshwater melt slows Atlantic meridional overturning circulation (Elliot et al., 2002; McManus et al., 2004; Oppo et al., 1995; Vidal et al., 1997), and this slowdown creates a nearly simultaneous change in sea surface temperatures in the Southern Hemisphere via the bipolar seesaw (Barker et al., 2009; Broecker 1998; 1986; Schmittner et al., 2002; Stocker and Johnson, 2003). In addition to an opposite change in sea surface temperature, there is evidence that changes in buoyancy (Watson and Garabato, 2006), latitudinal shifts in the

Westerlies (Anderson et al., 2009; Ninnermann and Charles, 1997; Toggweiler et al., 2006), and/or a changes in sea ice (Stephens and Keeling, 2000) affect the flow of $CO_2$ from the southern Ocean, which is an important reservoir for glacial/interglacial $CO_2$ (Knox and McElroy, 1984; Sarmiento and Togeweiler, 1984; Seigenthaler and Wenk, 1984; Anderson et al., 2009; Skinner et al., 2013).



Uncertainties about this general schema include questions about the role of changes in sea surface temperature relative to the location of the Westerlies/sea ice and the role of $CO_2$ from the Southern Ocean; does ventilation drive deglaciation or is it caused by the glaciation? Riveiros et al., (2013) postulate that termination V is driven "primarily via meridonal heat transport anomalies that would have enhanced the incipient warming arising from relatively weak

insolation forcing and only secondarily via $CO_2$ release." Conversely, Andersen et al., (2009) show that changes in the position of the Westerlies are the main driver for the increased flow of $CO_2$ to the atmosphere during the termination of the last ice age. Similarly, a shift by the Westerlies precedes the drop in atmospheric $CO_2$ during MIS 5 (Govin et al., 2009).

These competing hypothesis for terminations in general and stage 11 in particular can be tested by the statistical

ordering of the model errors. If changes in sea surface temperature initiate Termination V, the model's inability to simulate termination V will 'start' with its inability to simulate *SST*. This inability will be indicated by simulation errors for *SST* that precede and have information about the simulation errors for other variables. Specifically, simulation errors for other variables, such as $CO_2$, will not have prior information about the errors for *SST* and these errors will have prior information about the errors for the other variables, such as $CO_2$.

The statistical ordering of simulation errors indicates that the simulation errors for *SST* do not precede the model's inability to simulate MIS 11 and termination V. Errors for *SST* are preceded by the persisting errors for other variables (read across the *SST* row in Tables 3 and 4), such as $CO_2$, and the persisting errors for *SST* do not have prior information about the simulation errors for any variables (read down the *SST* column) at $p \le 0.05$. Using a threshold $p \le 0.10$, there is some evidence that persisting errors for *SST* have information about *Ice*. Consistent with these results,

conditioning the model on *SST*, which eliminates the simulation errors for *SST*, does not improve the model's ability to simulate *Ice* during MIS 11 relative to other potential causes for the stage 11 paradox (Figure 4). *In toto*, these results suggest that model failures do not 'start with' an inability to simulate sea surface temperature; rather the failure to simulate sea surface temperature is caused by the inability to simulate some other variable(s). As such, changes in sea surface temperature probably are not ultimately responsible for the 'stage 11 paradox.'

Instead, the statistical ordering generated by Equation (7) highlights the importance of the model's inability to simulate atmospheric carbon dioxide. Reading across the $CO_2$ row indicates that the simulation errors for other variables, including SST have no prior information about the simulation errors for $CO_2$, which suggests that model failures 'start with' an inability to simulate carbon dioxide. Furthermore, these failures propagate through the system. Reading down the $CO_2$ column indicates that the persisting errors for $CO_2$ have information about the simulation errors for other

variables including *Ice* and *SST* (Table 4).

The importance of carbon dioxide is consistent with results that indicate conditioning the model on observed values of $CO_2$ improves (compared to conditioning the model on *SST*) the model's ability to simulate *Ice* during MIS 11 (Figure 4). This supports the argument that high concentrations of $CO_2$ are responsible for the warm interglacial during MIS 11 (Yin and Berger, 2012). Together, these results suggest that terminations in general, and termination V in

particular, are driven by changes in atmospheric carbon dioxide. Furthermore, they are consistent with the notion that the peak in $CO_2$ concentrations drive changes in the glacial cycle that occur after 450 kyr BP (Pages, 2016). On the other hand, they contradict the notion that changes in carbon dioxide are a positive feedback loop in the Earth system, as opposed to a cause of glacial terminations (Ganopolski and Calov, 2011).



But the model's inability to simulate MIS stage 11 may not start solely with an inability to simulate $CO_2$. Persisting errors for *Ice* also are preceded by persisting errors for *Ca* and *Fe* (proxies for wind strength and aridity Section 2.1 and Supplemental Section I). And the persisting errors for *Ca* are preceded by the persisting errors for *Na* (a proxy for sea ice in the southern ocean Section 2.1 and Supplemental Section I) Although results cannot resolve the timing of
the model's inability to simulate wind (*Ca*, *Fe*) and sea ice (*Na*), their importance suggests that the model's inability to simulate the long interglacial of MIS 11 is generated in part by the model's inability to simulate the location and strength of winds, the extent of sea ice, and/or the ventilation of $CO_2$ from the Southern Ocean.

**5 Conclusion**

Our model is able to accurately simulate entire glacial cycles for an out-of-sample period that does not prescribe GHG forcing: the simulation is driven only by changes in orbital geometry. This ability suggests that the model can accurately hindcast climate using known climate parameters, which is the criterion proposed by Tzedakis et al., (2009) for understanding the current climate and where it is headed. Although satisfying this criterion has to be interpreted
with caution because predictability is not necessarily informative about the quality of a model with respect to capturing underlying causality (see e.g. Oreskes et al., 1994, or Clements and Hendry, 2005), the ability to hindcast climate suggests that our model could supplement the search for analogues for the Holocene (11,700 years before the present through the present), many of which focus on MIS 11 (Droxler et al., 2003; Tzedakis, 2010; Pol et al., 2011). Despite some similarities, our results suggest that such efforts are fraught with difficulty. Most importantly, the statistical
model cannot use the four measures of orbital geometry to simulate the depth and length of the interglacial that is associated with MIS 11. Conversely, the model is able to simulate many aspects of the current warm period (Figure 1 & 3): notable exceptions include peristing errors associated with *Ice* and *SST* (see below). This implies that any similarity in orbital geometry and feedback mechanisms (Imbrie et al., 1992; 1993, Ruddiman 2003; 2006) do not automatically translate into similar climates.  As such, there probably are important differences between the Holocene
and MIS 11.
Ironically, the interglacials during MIS 11 and the Holocene may share an important similarity: an important role for carbon dioxide. The inability to simulate the interglacial in MIS 11 is likely caused by a poorly-modelled physical mechanism that raises atmospheric carbon dioxide. It is highly unlikely that this mechanism is related to human activity, even though MIS 11 contains the first evidence for the use of fire by people in Britain (Gowlett, 2005; Preece
et al., 2006). Conversely, others argue that Holocene warming is amplified by anthropogenic emissions of carbon dioxide and methane (Ruddiman 2003; 2005; 2007).
Rather than trying to decide which aspects of the paleoclimate record 'line up' across marine isotope stages (e.g. Candy et al., 2014), future efforts will use the statistical model to identify the cause(s) for the current warming and how long it will last. Specifically, we will compile future values for orbital geometry and use them to simulate the
model as a CVAR-based alternative to GCM-based simulations (see e.g. Ganopolski et al., 2016). These CVAR simulations also will be used to assess the early Anthropogenic hypothesis by evaluating the degree to which anthropogenic emissions of carbon dioxide and methane can account for outliers and persisting errors in *Ice* and other climate variables during the Holocene.



**Data and Code Availability:** The data and computer code used in this analysis are available on OpenBU, which is FAIR-compliant, and can be accessed through a globally unique and eternally persistent identifier, https://hdl.handle.net/2144/40340 P. This dataset is distributed under the terms of the Creative Commons Attribution-ShareAlike 4.0 License (http://creativecommons.org/licenses/by-sa/4.0).

**Team list:** The team includes Robert Kaufmann (RK) and Felix Pretis (FP).

**Author contributions:** This project was conceived by RK and FP. RK compiled the data from the statistical model
and FP did the statistical analysis to identify impulses and steps. RK and FP write the manuscript, designed the tables, and created the figures together.

**Competing Interests:** The authors have no financial or non-financial interests associated with the material in this manuscript.

**Acknowledgements:** We thank David F Hendry, Luke Jackson, and Katarina Juselius for helpful comments and suggestions. Financial support from the Robertson Foundation and British Academy is gratefully acknowledged."

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



**Figure captions**

Fig. 1. The observed values for temperature (black line) and values simulated by the system model conditioned only on the four variables for solar insolation (red line). Thick portions of the red line represent time steps in which the simulation error is significantly different from zero (non-zero error). Red circles represent time steps when the simulation error is an (innovational) outlier. The light gray area is the out-of-sample forecast period; MIS 11 is shaded dark gray. (b) same as above for carbon dioxide, (c) same as above for methane, (d) same as above for land ice, (e) same as above for $Na$, (f) same as above for $SO_4$, (g) same as above for sea level, (h) same as above for $SST$[1].

Fig. 2. The root mean square simulation errors for all ten endogenous climate variables simulated by the CVAR system model conditioned only on the four variables for solar insolation. RMSE for the in-sample period are shown as dark grey (left), out-of sample as grey (middle), and out-of sample excluding Marine Isotope Stage 11 as light grey (right).

Fig. 3. The number of outliers (red spikes) and non-zero errors (darkly shaded) for each time step. Marine isotope stages are indicated by alternating areas of shading.

Fig. 4: The value of $Ice$ conditioned on orbital geometry only (green line), $CO_2$ (red line), non-sea-salt sodium (blue line), $Ca$ (yellow line), and $SST$ (orange line). Values from the proxy record are given by the black line.

---

[1] Note that the series of SST exhibits non-zero simulation errors nearly throughout the sample, suggesting a non-zero bias throughout the observational record – simulated model values persistently exceed observations.





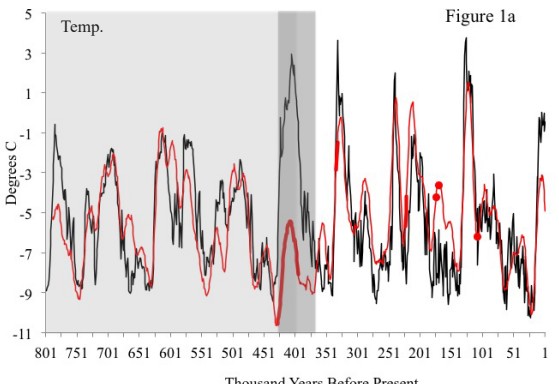

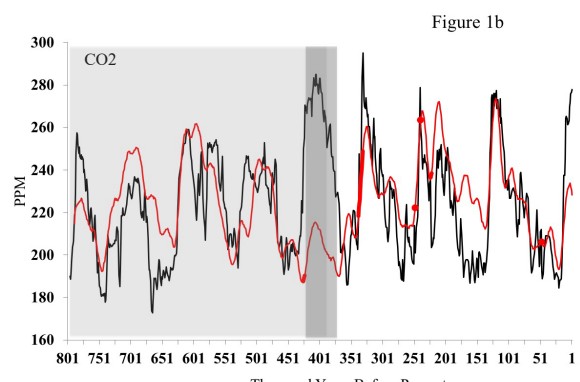

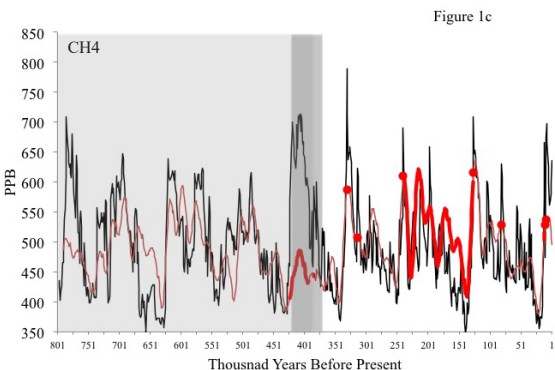

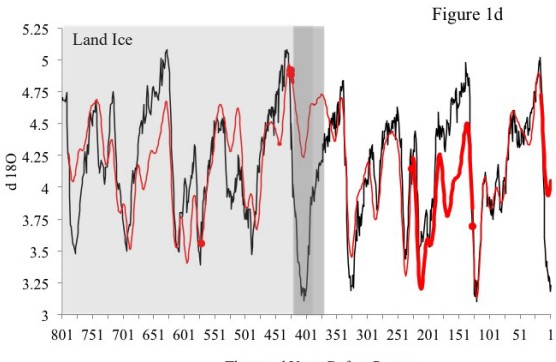

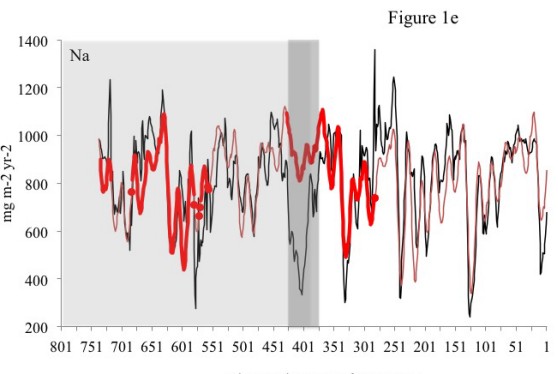

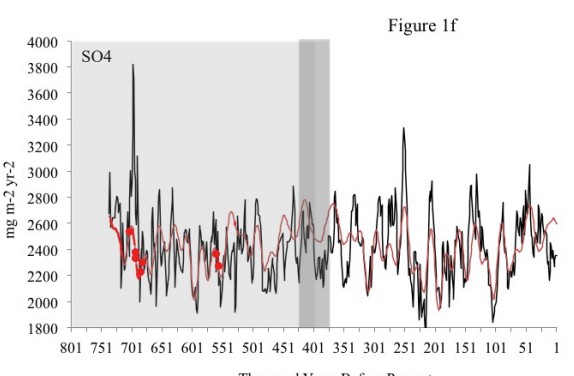

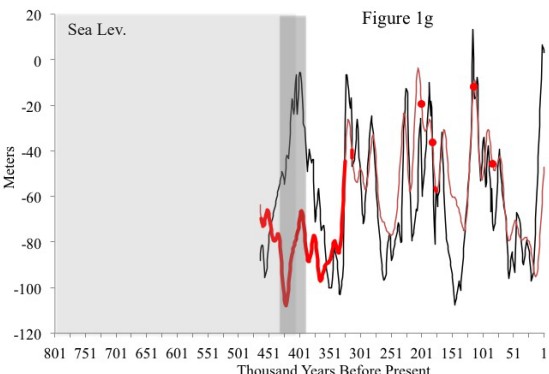

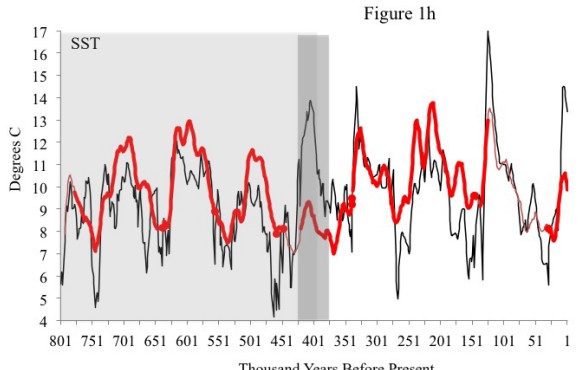





Figure 2

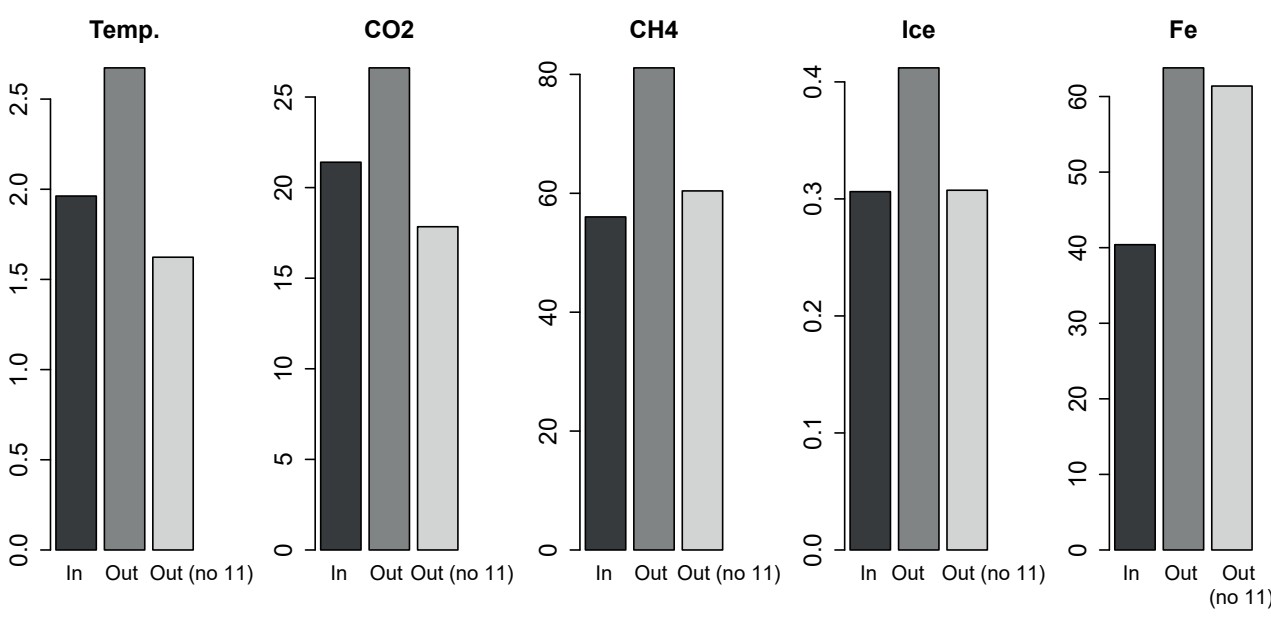

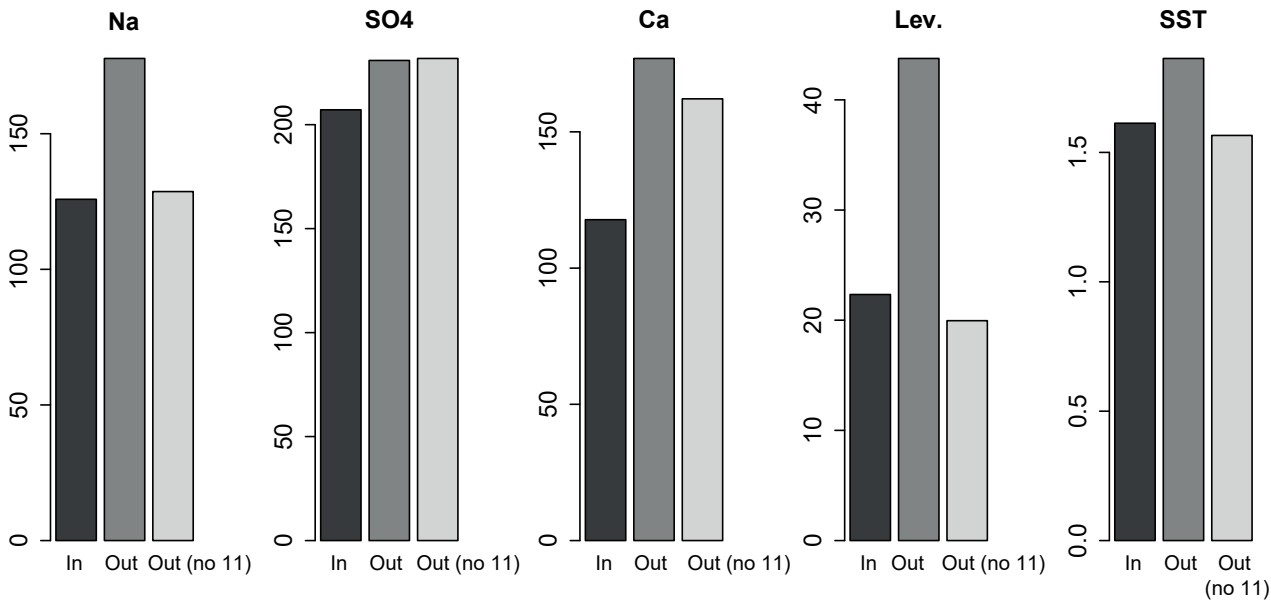



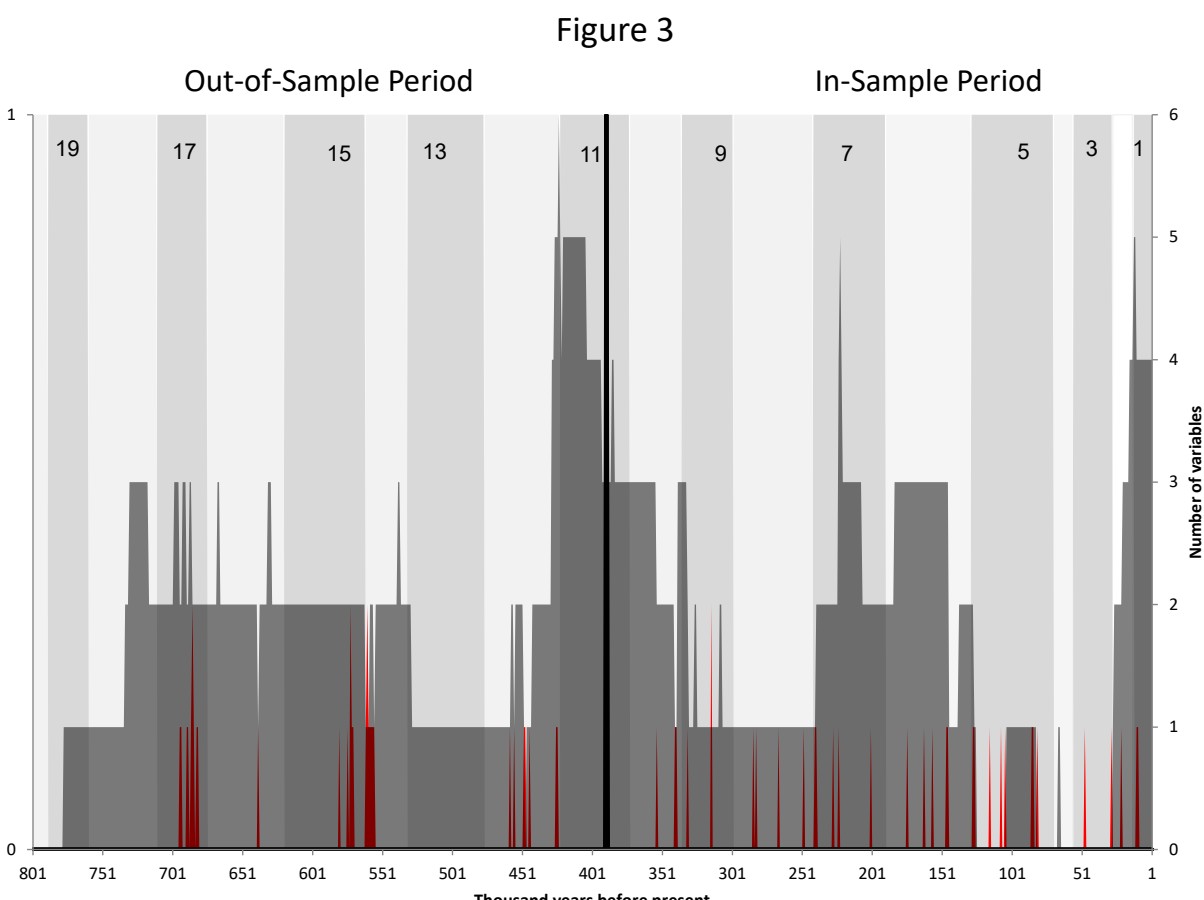

Figure 3



Figure 4






**Table 1:** Time series included in the CVAR

| Variable | Source | Unit | Time Scale | Obs | First Observation |
|---|---|---|---|---|---|
| Temp | Jouzel *et al.*, (2007) | Δ avg. last 1 kyr | EDC3 | 710 | 801kyr BP |
| $CO_2$ | Lüthi *et al.*, 2008 | ppmv | ECD3 | 517 | 798 kyr BP |
| $CH_4$ | Loulergue *et al.*, (2008) | ppbv | EDC3 | 1477 | 799 kyr BP |
| Ice | Lisiecki and Raymo, (2005) | $\delta180$ | LR04 | 390 | 801kyr BP |
| Fe | Wolff *et al.* (2006) | $\mu g\ m^{-2}yr^{-1}$ | EDC2 | 187 | 736 kyr BP |
| Na | Wolff *et al.* (2006) | $\mu g\ m^{-2}yr^{-1}$ | EDC2 | 195 | 739 kyr BP |
| SO4 | Wolff *et al.* (2006) | $\mu g\ m^{-2}yr^{-1}$ | EDC2 | 195 | 739 kyr BP |
| Ca | Wolff *et al.* (2006) | $\mu g\ m^{-2}yr^{-1}$ | EDC2 | 195 | 739 kyr BP |
| Sea Level | Siddal *et al.*, (2003) | Meters | SPECMAP | 125 | 466 kyr BP |
| Sea Surface Temp | Martinez-Garcia *et al.*, (2009) | Degrees C | EDC3 | 121 | 801kyr BP |
| Eccentricity | Paillard *et al.*, (1996) | Dimensionless index | _ | 801 | 801kyr BP |
| Obliquity | Paillard *et al.*, (1996) | Degrees | _ | 801 | 801kyr BP |
| Precession | Paillard *et al.*, (1996) | Dimensionless index | _ | 801 | 801kyr BP |
| Seasonal Insolation | Paillard *et al.*, (1996) | $W/m^2$ | _ | 801 | 801kyr BP |



**Table 2:** Tests of simulation accuracy during various periods.

| | In vs. out-of-sample | | Distribution among marine isotope stages | | | |
| --- | --- | --- | --- | --- | --- | --- |
| | | | Outliers | | Persisting errors | |
| Variable | Outliers | Persisting errors | All stages | Stage 11 excluded | All stages | Stage 11 excluded |
| Temp | $3.0^{+}$[0/3] | $19.8^{**}$[34/6] | 18.4 | 15.6 | $286.5^{**}$ | $53.8^{**}$ |
| $CO_2$ | $3.0^{++}$[0/3] | 2.5 [4/10] | 15.8 | 13.2 | $57.9^{**}$ | $53.3^{**}$ |
| $CH_4$ | $7.0^{**+}$[0/7] | $49.3^{**}$[31/117] | $46.5^{**}$ | $39.7^{**}$ | $535.9^{**}$ | $571.8^{**}$ |
| Ice | 0.2 [3/2] | $103.7^{**}$[4/116] | 14.9 | 3.8 | $543.0^{**}$ | $506.5^{**}$ |
| Fe | 1.4 [1/4] | $91.8^{**}$[226/79] | 19.1 | 12.6 | $325.4^{**}$ | $313.5^{**}$ |
| Na | $4.2^{*+}$[6/1] | $24.2^{**}$[172/107] | 28.2 | 3.5 | $356.0^{**}$ | $342.2^{**}$ |
| SO4 | $7.9^{**+}$[7/0] | $40.7^{**}$[36/0] | $80.2^{**}$ | 3.8 | $398.2^{**}$ | $368.7^{**}$ |
| Ca | 1.3 [3/1] | $24.3^{**}$[35/6] | 22.9 | 3.00 | $332.3^{**}$ | $307.0^{**}$ |
| Sea Level | 0.8 [0/4] | $136.3^{**}$[73/69] | 8.0 | 6.3 | $288.1^{**}$ | $275.4^{**}$ |
| SST | 0.1 [4/5] | $9.1^{**}$[357/282] | 20.0 | 11.6 | $101.7^{**}$ | $101.6^{**}$ |
| All | 0.1 [24/30] | $52.5^{**}$[972/793] | $37.9^{**}$ | 3.5 | $561.4^{**}$ | $427.7^{**}$ |

Value rejects the null hypothesis at p < .05 ($^{*}$), p < 0.01 respectively ($^{**}$). Blue indicates the out-of-sample simulation is more accurate than the in-sample simulation; red indicate the in-sample simulation is more accurate. Values in brackets indicate the number of outlier/persisting errors in the out-of-sample period relative to the number of outlier/nonzero mean errors in the in-sample period.. A large value implies that the in-sample simulation has significantly fewer outlier/persisting errors, which would make it more accurate than the out-of-sample simulation.





**Table 3:** p-values for test of significance on $\theta_j$ (equation 8) for the sample that includes all endogenous variables. Red indicates rejection of the exclusion ($p < 0.05$) of lagged errors of other series, blue indicates rejection of the exclusion ($p < 0.10$) of lagged errors of other series, and green indicates rejection of the exclusion for the autoregressive lags ($p < 0.05$). The red value of 0.023 in the second column of the first row indicates that the indicates that the simulation errors for CO2 have information about the simulation errors for temperature.

| Dep. Variable Eq. 8 | Non zero simulation error excluded from equation 8 | | | | | | | | | |
|---|---|---|---|---|---|---|---|---|---|---|
| | Temp | CO₂ | CH₄ | Ice | Fe | Na | SO4 | Ca | Level | SST |
| Temp | 0.005 | 0.023 | 0.041 | 0.144 | 0.140 | 0.314 | 0.314 | 0.878 | 0.795 | 0.169 |
| CO₂ | 0.397 | 0.629 | 0.057 | 0.143 | 0.721 | 0.760 | 0.760 | 0.676 | 0.512 | 0.992 |
| CH₄ | 0.829 | 0.516 | 0.000 | 0.074 | 0.285 | 0.101 | 0.101 | 0.168 | 0.746 | 0.867 |
| Ice | 0.421 | 0.148 | 0.496 | 0.277 | 0.270 | 0.334 | 0.334 | 0.393 | 0.272 | 0.051 |
| Fe | 0.054 | 0.055 | 0.658 | 0.586 | 0.014 | 0.910 | 0.910 | 0.000 | 0.337 | 0.600 |
| Na | 0.013 | 0.442 | 0.064 | 0.917 | 0.007 | 0.875 | 0.875 | 0.752 | 0.476 | 0.622 |
| SO4 | 0.234 | 0.283 | 0.111 | 0.705 | 0.301 | 0.902 | 0.902 | 0.042 | 0.957 | 0.158 |
| Ca | 0.044 | 0.842 | 0.884 | 0.902 | 0.032 | 0.475 | 0.475 | 0.006 | 0.965 | 0.106 |
| Level | 0.259 | 0.152 | 0.422 | 0.028 | 0.405 | 0.415 | 0.415 | 0.938 | 0.637 | 0.617 |
| SST | 0.015 | 0.036 | 0.368 | 0.052 | 0.949 | 0.119 | 0.119 | 0.271 | 0.969 | 0.000 |





**Table 4:** p-values for test of significance on $\theta_j$ (equation 8) for the sample that includes variables other than sea-level. Red indicates rejection of the exclusion ($p < 0.05$) of lagged errors of other series, blue indicates rejection of the exclusion ($p < 0.10$) of lagged errors of other series, green indicates rejection of the exclusion for the autoregressive lags.

| Dependent variable Eq. 8 | Non zero simulation error excluded from equation 8 | | | | | | | | |
|---|---|---|---|---|---|---|---|---|---|
| | Temp | $CO_2$ | $CH_4$ | Ice | Fe | Na | $SO_4$ | Ca | SST |
| Temp | 0.009 | 0.311 | 0.128 | 0.349 | 0.173 | 0.644 | 0.890 | 0.571 | 0.187 |
| $CO_2$ | 0.186 | 0.852 | 0.213 | 0.136 | 0.700 | 0.312 | 0.866 | 0.149 | 0.847 |
| $CH_4$ | 0.454 | 0.471 | 0.362 | 0.371 | 0.568 | 0.181 | 0.878 | 0.775 | 0.886 |
| Ice | 0.314 | 0.006 | 0.584 | 0.000 | 0.035 | 0.228 | 0.370 | 0.046 | 0.055 |
| Fe | 0.189 | 0.202 | 0.394 | 0.860 | 0.064 | 0.711 | 0.976 | 0.059 | 0.791 |
| Na | 0.212 | 0.668 | 0.632 | 0.881 | 0.272 | 0.812 | 0.066 | 0.687 | 0.932 |
| $SO_4$ | 0.166 | 0.226 | 0.039 | 0.881 | 0.736 | 0.460 | 0.000 | 0.025 | 0.246 |
| Ca | 0.094 | 0.954 | 0.766 | 0.955 | 0.376 | 0.024 | 0.000 | 0.113 | 0.204 |
| SST | 0.714 | 0.011 | 0.326 | 0.012 | 0.973 | 0.337 | 0.715 | 0.180 | 0.133 |