# Peer review of "Testing Hypotheses About Glacial Dynamics and the Stage 11 Paradox Using a Statistical Model of Paleo-Climate"

_Climate of the Past, 2020_

## Referee Comment (RC1) · Anonymous Referee #1 · 8 Jun 2020

The authors employ the model of Kaufmann and Juselius (2013) (KJ2013) fitted with data to 391 kyr BP to simulate data up to 791 kyr BP. The main idea is to test for structural breaks by comparing the in-sample and out-of-sample fit of the simulated data. If the fits are quite different, there may have been a structural break. The main finding is that the simulated data based on the model fit using data after the mid-Brunhes event fits the observed data well prior to the event, which suggests that there is not a major structural break at that time. My overall impression is that there is a really interesting idea here. I reviewed an earlier version of this manuscript for another journal in November 2016. The authors were very responsive to my comments. I reviewed a revision for the same journal in May 2017 and recommended a conditional

acceptance to that journal pending some very minor revisions. Although the manuscript has changed since 2017 (including the title), my comments raised here are almost the same as those raised then, adjusted for formatting, because the typos I noticed then have still not been fixed.

Comments

1. I'm not sure if it is a requirement of the journal, but the paragraph formatting (no indent, no additional spacing) makes the manuscript hard to read. 2. Pg. 3, line 7: I can't tell whether it is just one linea are all lines afterward, but the alignment is definitely off here. 3. Pg. 3, line 10: "five sections" – just a suggestion, I would write "next four sections" 4. Pg. 4, equation 2: no transpose after $z_{t-1}$ 5. Pg. 4, line 35: z should be $z_{t}$ (add subscript t) 6. Pg. 5, line 2: "identicially" should be "identically" 7. Pg. 5, line 10: something is off with the spacing around $\alpha$ and $\beta$ 8. Pg. 5, line 14: "model model" repeated word 9. Pg. 5, line 24: something looks wrong with the mathematical expression in this line... isn't $\epsilon_{t}=0$? 10. Pg. 7, equation 7: $\omega$ should be w 11. Pg. 7, line 19: (s) should be (S) – upper case to be consistent with equation 7 12. Pg. 11, line 14: "meridional" is correct, but I found five instances of misspelled "meridonal" in the paper 13. Supplement, equation 6: A few things off here... two equals signs? Shouldn't the index in the denominator be j and not t? 14. Supplement, footnote 2: "$\mu_j=0$ implies..." – need something like "for all j" 15. Please proofread and spellcheck everything!

---

## Referee Comment (RC2) · Anonymous Referee #2 · 11 Jun 2020

The authors use a statistical model to fit time series with a 1kyr resolution, of 10 proxy records (temperature, CO2, methane, SST, land ice volume, sea level, Fe, Na, SO4, etc) of the ice ages over the past 391 kyr, and then to simulate the past 791 kyr. The forcings are prescribed orbital parameter time series. Their conclusions are that nonlinearities are not important, and they discuss at some length the model failure to simulate the period around 400 kyr. If I understand correctly equation (2) and the following text, the model has a \*huge\* number of parameters: there are several 10x14 and 10x15 matrices of model parameters, involving hundreds of parameters being fit. If this is indeed the case, I don't know that it is a surprise or an achievement that the model achieves a satisfactory fit. Saltzman (not cited by the authors) developed his

"Earth system models" with the philosophy that the goal of glacial models is to produce a good fit to the ice volume record with the smallest number of parameters, although it is not clear if one actually learned about the dynamics of ice ages by such a fit. In any case, he used a nonlinear model and the order of 10-15 parameters, which, if I understand correctly, is much less than is used here. I can see some (vague) similarity between the Saltzman philosophy and the one taken here, in the sense that the authors try to fit the record without suggesting a mechanistic understanding of ice ages. This could still have led to a useful insight if they could indeed show that nonlinearity is not important, although as I discuss below I don't believe they have shown that.

First a comment on the data: while ten proxy records are being used (the authors should plot them), they are likely not very independent, as they tend to mostly vary together with each other, so the number of observations being explained/ fitted is not as large as it might seem superficially, making the number of model parameters effectively even larger.

The authors emphasize as their main conclusion that their study rules out previous claims that nonlinearity must be important. This would have been novel and interesting, but I don't find this convincing for two reasons. (1) Their model is somewhat nonlinear. They need to completely linearize it, repeat the analysis, and demonstrate that the results and conclusions are robust. (2) Given the very large number of model parameters, I suspect their approach could fit any low-order *nonlinear* dynamics successfully. To make a satisfactory case, they need to test this hypothesis as follows: build a simple nonlinear model based on 10 weakly coupled nonlinear oscillators (e.g., Van der pol oscillators, see Crucifix 2013 "Why could ice ages be unpredictable?" Clim. Past, 9, 2253-2267); fit a similar stochastic model to this model output. The null hypothesis could be that such a model output would be possible to fit using the nearly-linear model used by the authors, although the dynamics are clearly strongly nonlinear. If the null hypothesis is not satisfied, the authors would have a stronger case.

Page 2, line 15-20: That $CO_2$ is not an external variable seems obvious. I don't know

that this adds anything new to our understanding.

Page 3, lines 5-10: it seems less plausible that glacial terminations are driven by atmospheric or oceanic dynamics. The trigger needs to be a change to a climate system component that has a long 90 kyr time scale, which reaches some critical threshold, starts changing, and that then leads to changes in the faster components such as sea ice, AMOC, atmospheric circulation, etc. The only such slow component is land ice, which may take 90 kyr to reach some critical size that then affects other components. I realize there is much in the literature about the AMOC triggering terminations etc, but the above argument seems to suggest that these ideas are not likely to be realistic.

Page 8, lines 10-15: it seems to me that the model's ability to simulate glacial cycles during the out-of-sample period does not mean the model is correct. It just seems to suggest that the dynamics of the glacial cycle are the same throughout the past 800 kyr. It seems still possible that the model simply fits the record due to its very large number of parameters.

Because of the issues mentioned above, it seems that me that the manuscript as it stands now does not make a strong case for the suggested conclusions. I recommend a major revision, that it my opinion needs to require new analysis and results rather than just a rewrite and further explanations. I hope the authors find these comments are helpful.

---

## Author Comment (AC1) · 9 Jul 2020

We thank the reviewer for (again) reviewing the manuscript. We are gratified to read that he/she writes that "My overall impression is that there is a really interesting idea here." Given the reviewer's history with the manuscript, we hope that they will read about the results using the van-der-Pol oscillator in the revised manuscript; it greatly adds to the interesting idea. And of course, we will correct all of the typos identified by the reviewer.

―――――――――――――――――

---

## Author Comment (AC4) · 14 Jul 2020

We thank the reviewer for their careful review of the manuscript and detailed comments. Based on these comments, we should have made clear that the CVAR model does not over-fit the data and how it copes with non-linearities. These issues are described in a point-by-point response.

*************************** Reviewer Comment ******************************

If I understand correctly equation (2) and the following text, the model has a *huge* number of parameters: there are several 10x14and 10x15 matrices of model param-

eters, involving hundreds of parameters being fit. If this is indeed the case, I don't know that it is a surprise or an achievement that the model achieves a satisfactory fit. Saltzman (not cited by the authors) developed hisC1"Earth system models" with the philosophy that the goal of glacial models is to produce a good fit to the ice volume record with the smallest number of parameters, although it is not clear if one actually learned about the dynamics of ice ages by such a fit. In any case, he used a nonlinear model and the order of 10-15 parameters, which, if I understand correctly, is much less than is used here. I can see some (vague) similarity between the Saltzman philosophy and the one taken here, in the sense that the authors try to fit the record without suggesting a mechanistic understanding of ice ages. This could still have led to a useful insight if they could indeed show that nonlinearity is not important, although as I discuss below I don't believe they have shown that.
* * *
The current version of the manuscript does not address the reviewer's concern regarding the number of estimated parameters. We will address this in our revised manuscript. The literature has published thousands of papers that estimate CVAR models; many are estimated using far fewer observations than available here (such as 50 observations per time series as commonly found in macroeconomics). As such, the sample of about 400 observations per time series is comparably large.

In statistics, the potential pitfalls of a statistical model over-fitting the data is captured by the degrees of freedom, the number of independent pieces of information to estimate another piece of information. To illustrate, a line [Y(t)= $\alpha+\beta$X(t)+$\mu$(t)] will perfectly fit two observations, but the model has zero degrees of freedom and therefore is not statistically meaningful. But fitting that line to one hundred observations would have 98 degrees of freedom and so could represent a statistically meaningful relation.

For the CVAR reported by Kaufmann and Juselius (2013), each dependent variable has 390 observations. The right-hand side specifies 33 variables, which leaves 357

degrees of freedom for each equation. This is a very large number. Empirical statistical results are evaluated against tables that list critical values. The average table used to evaluate t statistics reports values for up to 100 degrees of freedom, 250 degrees of freedom, and then the asymptotic value for an infinite number of degrees of freedom. This implies that a sample of 357 observations falls near the 'top end' of observations used in empirical investigations.

Furthermore, many of the equations contain more than 357 degrees of freedom due to restrictions placed on the estimated coefficients. To identify the system, coefficients on many variables are restricted to zero. For example, the first cointegrating relation in Table 2 of Kaufmann and Juselius, (2013) and Supplementary Table S.1 restricts thirteen variables (other than Temp and CO2) to zero, which increases the degrees of freedom by thirteen. Similar restrictions are imposed on the other cointegrating relations such that of the possible 140 parameters in the 10x14 long-run $\Pi$ matrix, 106 parameters are restricted to be zero, which indicates that only 24 parameters are actually estimated (and these restrictions are not rejected using likelihood ratio tests). As such, there are more than 357 degrees of freedom in each equation.

Finally, the model is used to generate an out-of-sample forecast. If the CVAR model overfit the in-sample observations, it is highly unlikely (from a statistical perspective) that it would be able to fit the out-of-sample period with about equal accuracy. As such, the accuracy of the out-of-sample period is consistent with the hypothesis that the large number of degrees of freedom minimizes concerns about overfitting.

Thus, over-fitting should not be a major concern, and also does not seem to occur as we show in the example using a van-der-Pol oscillator following the reviewer's suggestion below. We will clarify these issues in the revised version of the manuscript and include references to the work of Saltzman

************************* Reviewer Comment ****************************

First a comment on the data: while ten proxy records are being used (the authors

should plot them), they are likely not very independent, as they tend to mostly vary together with each other, so the number of observations being explained/ fitted is not as large as it might seem superficially, making the number of model parameters effectively even larger.

\*\*\*\*\*\*\*\*\*\*\*\*\*\*\*\*\*\*\*\*\*\*\*\*\*\*\*\*\*\*\*\*\*\*\*\*\*\*\*\*\*\*\*\*\*\*\*\*\*\*\*\*\*\*\*\*\*\*\*\*\*\*\*\*\*\*\*\*\*\*\*\*\*\*\*\*\*\*

One of the reasons that we use the CVAR model to analyse the paleoclimate data is that is solves many of the difficulties associated with traditional regression techniques. These advantages are described in Juselius (2014) who writes, "By exploiting the unit root feature, typical of many economic variables [and other non-stationary time series such as climate data], the CVAR model was shown to solve the problem of (1) time dependent residuals by conditioning on sufficiently many lags and controlling for a changing environment when needed, (2) spurious correlation and regression results, (3) multicollinearity [correlation among the proxy records in our case], (4) normalization, and (5) reduced rank." We will add this information to the revised manuscript. As such, collinearity among the climate variables should not affect the statistical results beyond increasing the estimates of the error variance which is explicitly accounted for in our statistical tests.

\*\*\*\*\*\*\*\*\*\*\*\*\*\*\*\*\*\*\*\*\*\*\*\*\*\*\* Reviewer Comment \*\*\*\*\*\*\*\*\*\*\*\*\*\*\*\*\*\*\*\*\*\*\*\*\*\*\*\*

The authors emphasize as their main conclusion that their study rules out previous claims that nonlinearity must be important. This would have been novel and interesting, but I don't find this convincing for two reasons. (1) Their model is somewhat nonlinear. They need to completely linearize it, repeat the analysis, and demonstrate that the results and conclusions are robust.

\*\*\*\*\*\*\*\*\*\*\*\*\*\*\*\*\*\*\*\*\*\*\*\*\*\*\*\*\*\*\*\*\*\*\*\*\*\*\*\*\*\*\*\*\*\*\*\*\*\*\*\*\*\*\*\*\*\*\*\*\*\*\*\*\*\*\*\*\*\*\*\*\*\*\*\*\*\*

We apologize; we should have been clearer regarding the linear nature of the CVAR model. The CVAR model is linear in parameters. That is, the first cointegrating relation

in Table 2 of Kaufmann and Juselius (2013) and Supplementary Table S.1 is a linear relation between Temp and CO2 and the third cointegrating relation is a linear relation among Ice, CO2, and eccentricity. These linear long-run cointegrating relations, which lie at the heart of the CVAR, is what we refer to as a 'linear model'. We will clarify this in our revised manuscript.

The apparent non-linear dynamics can stem from two sources. First, the individual variables (specifically, the first difference of each variable) such as temperature adjust towards disequilibrium as a linear function of disequilibrium in the level of the variables in the previous time period. This creates a seeming non-linear change in the level. However, the model is linear in both first differences and levels.

Second, the model is conditioned on orbital geometry, which changes nonlinearly over time. But these nonlinear changes are represented linearly. As such, non-linear changes in orbital geometry have a linear relation with the variables simulated by the model. This linear effect is very different than the nonlinearities and/or threshold effects that are described in the literature that we cite.

************************* Reviewer Comment *****************************

(2) Given the very large number of model parameters, I suspect their approach could fit any low-order *nonlinear* dynamics successfully. To make a satisfactory case, they need to test this hypothesis as follows: build a simple nonlinear model based on 10 weakly coupled nonlinear oscillators (e.g.,Van der pol oscillators, see Crucifix 2013 "Why could ice ages be unpredictable?" Clim.Past, 9, 2253-2267); fit a similar stochastic model to this model output. The null hypothesis could be that such a model output would be possible to fit using the nearly-linear model used by the authors, although the dynamics are clearly strongly nonlinear. If the null hypothesis is not satisfied, the authors would have a stronger case.
* * *
Thank you, the reviewer proposes a very interesting test of our model. Create observations using a nonlinear data generating process, in this case, a van-der-Pol oscillator from Crucifix (2013). Then fit a linear statistical model to the data. If the linear statistical model can fit the data well, this would suggest that a linear model can simulate the nonlinear dynamics used to generate the data. Such a result would weaken our claim that nonlinearities/threshold effects do not play an important role in glacial cycles. Conversely, if the model fails to fit the data well, this would indicate that the linear statistical model cannot the nonlinear dynamics used to generate the data. This result would support our claim because our linear statistical model fits the in- and out-of-sample observations for paleoclimate well.

We test the reviewer's hypothesis by using the van-der-Pol oscillator from Crucifix (2013) to generate two non-linear sets of data. First, we construct a two variable van-der-Pol oscillator in discrete time that is conditioned on a sinusoidal forcing F (similar to Crucifix, 2013), which is perturbed with white noise. The artificial van-der-Pol data is shown in Figure 1. The variable 'y' mimics a suddenly changing time series in the paleo-proxy record (such as temperatures) while 'x' mimics a more gradually changing variable (such as ice).

We repeat this exercise using a ten variable van-der-Pol oscillator. The ten variable oscillator is specified to simulate one variable that changes suddenly and nine variables that accumulate gradually. The ten variable system is shown in Figure 2.

For each of these simulations, we use half of the simulated data (area shaded in grey) to fit an 'in-sample' linear vector autoregression (VAR) model (See Supplemental Material) in which Yt is a vector of two (y and x; or ten in the case of the larger system) variables generated by the van-der-Pol oscillator, F is the sinusoidal forcing, (s) is the number of lags (s) chosen using the Schwarz information criterion (Schwarz, 1978), and Ït_t is a vector of error terms. The VAR corresponds to the CVAR model by Kaufmann and Juselius (2013). We use the statistical model to simulate the endogenous variables over the full sample, which mirrors the approach used in our manuscript.

Results from simulating van-der-Pol Oscillators

Visual inspection of the simulation in Figures 1 and 2 indicate that the linear system does not match the abrupt non-linear pattern. The red lines in Figures 1 and 2 indicate that the model does not account for much of the variation during the in-sample period. Nor does this performance improve during the out-of-sample period. The same holds true for both in- and out-of-sample generated by the ten-variable oscillator (see Figure 2).

This visual impression is confirmed statistically by testing whether the model errors, (the difference between the black and red lines in figures 1) are statistically different from zero. We use the same indicator saturation technique, which is used in the manuscript, to identify periods when the model errors are statistically different from zero for two or more consecutive periods (steps). As indicated by the steps in Figure 3, model errors are statistically different from zero for most of the sample period in the two variable case (Figure 3).

These simulation results show that a linear VAR model is unable to successfully simulate a non-linear van-der-Pol system. Conversely, the linear CVAR climate model is able to simulate glacial dynamics, as described in our manuscript. Together, these results suggest that non-linear dynamics (such as in the van-der-Pol system) may not be play a large role in glacial cycles. We thank the reviewer for this clever test and will include it in our revised manuscript.

*************************** Reviewer Comment ******************************

Page 2, line 15-20: That $CO_2$ is not an external variable seems obvious. I don't know that this adds anything new to our understanding.
* * *
We included this note because there is a mismatch between the physical climate system and many of the models used to simulate it. As indicated by the references in

our manuscript, the atmospheric concentration of CO2 is endogenous; it is driven by temperature, sea ice, and many other variables. But many of these mechanisms are not simulated by existing empirical and process-based models of the climate system. Without the ability to simulate these mechanisms, changes in the atmospheric CO2 over time are treated as an exogenous variable. For example, the ice model CLIMBER 2 is conditioned on the radiative forcing of CO2 (Ganopolski and Calov, 2011). We will clarify this in the revised manuscript.

*************************** Reviewer Comment ******************************

Page 3, lines 5-10: it seems less plausible that glacial terminations are driven by atmospheric or oceanic dynamics. The trigger needs to be a change to a climate system component that has a long 90 kyr time scale, which reaches some critical threshold, starts changing, and that then leads to changes in the faster components such as sea ice, AMOC, atmospheric circulation, etc. The only such slow component is land ice, which may take 90 kyr to reach some critical size that then affects other components. I realize there is much in the literature about the AMOC triggering terminations etc, but the above argument seems to suggest that these ideas are not likely to be realistic.
* * *
We respectfully disagree with the heart of reviewer's comment "The trigger needs to be a change to a climate system component that has a long 90 kyr time scale, which reaches some critical threshold, starts changing, and that then leads to changes in the faster components such as sea ice, AMOC, atmospheric circulation, etc." One important point of our paper is that there is no need to invoke this nonlinear threshold effect. The linear CVAR model is able to accurately simulate glacial cycles both in and out of sample. As indicated by the 'van-der-Pol oscillator experiment' suggested by the reviewer, it is highly unlikely that the linear CVAR model would be able to simulate glacial cycles in and out of sample if the paleoclimate data were dominated by a non-linear data generating process. We recognize that this may be a different and somewhat con-

troversial way of looking at the data, but we think that the manuscript and the results of the van-der-Pol oscillator experiment suggested by the reviewer represents sound scientific evidence that the community should consider.

************************* Reviewer Comment ******************************

Page 8, lines 10-15: it seems to me that the model's ability to simulate glacial cycles during the out-of-sample period does not mean the model is correct. It just seems to suggest that the dynamics of the glacial cycle are the same throughout the past 800kyr. It seems still possible that the model simply fits the record due to its very large number of parameters.
* * *
We agree scientists can never be sure that a model is correct and that the in- and out-of-sample primarily suggests stability of the processes. However, we hope that the simulation exercise using the van-der-Pol oscillator (see reply above) strengthens our argument.

************************* Reviewer Comment ******************************

Because of the issues mentioned above, it seems that me that the manuscript as it stands now does not make a strong case for the suggested conclusions. I recommend a major revision, that it my opinion needs to require new analysis and results rather than just a rewrite and further explanations. I hope the authors find these comments are helpful.
* * *
Literature Cited

Crucifix, M.: Why could ice ages be unpredictable? Climate of the Past, 9, 2253–2267, doi:10.5194/cp922532013, 2013. Ganopolski, A., and Calov, R.: The role of orbital forcing, carbon dioxide, and regolith in 100 kyr glacial cycles, Climate of the

Past, 7:1415-1425, 2011. Juselius, K.: Haavelmo's probability approach and the cointegrated VAR, Econometric Theory, 31:213-232, 2015. Kaufmann, R.K., and Juselius, K.: Testing hypotheses about glacial cycles against the observational record, Paleoceanography 28, 1–10, doi:10.1002/palo.20021, 2013, Saltzman, B., Maasch, K. A., & Verbitsky, M. Y.: Possible effects of anthropogenically‐increased CO2 on the dynamics of climate: Implications for ice age cycles. Geophysical Research Letters, 20(11), 1051-1054, 1993. Schwarz, G.: Estimating the dimension of a model, Annals of Statistics 6: 461-464, 1978.

Please also note the supplement to this comment:
https://cp.copernicus.org/preprints/cp-2020-58/cp-2020-58-AC4-supplement.pdf

———————————————

[Figure]

[Figure]

**van−der−Pol: Data and Model Simulation**

Fig. 1. Simulating artificial data generated by a two-variable van-der-Pol Oscillator (Crucifix 2013). Grey shows artificial time series y and x driven by the exogenous sinusoidal forcing F. Red shows the sim

[Figure]

**Fig. 2.** Simulating artificial data generated by a ten-variable van-der-Pol Oscillator (Crucifix, 2013). Grey shows artificial time series y and x driven by the exogenous sinusoidal forcing F. Red shows the si

**van−der−Pol: Model Simulation Errors**

In−Sample    Out−of−Sample

Model Error: y

In−Sample    Out−of−Sample

Model Error: y

**Fig. 3.** Simulation errors when modelling the non-linear van-der-Pol oscillator using the linear VAR model. Red shows the simulation errors of y and x, blue shows the time-varying mean of the simulation errors

[Figure]

**Supplement:**

$$Y_t = \alpha + \sum_{i=1}^{s_1} A_i Y_{t-i} + \sum_{i=0}^{s_2} \phi_i F_{t-i} + \epsilon_t$$

---

## Referee Comment (RC3) · Anonymous Referee #3 · 16 Jul 2020

This contribution is based on a statistical model called CVAR, standing for "cointegration vector autoregression".  The model is calibrated on the latest four glacial-interglacial cycles, and then shown to fully reproduce the sequence of the latest eight glacial-interglacial cycles, except for the deglaciation leading to the stage 11. The authors conclude that this deglaciation, associated with the mid-Brunhes event, is to be considered as an anomalous event rather than marking a regime transition. By inspecting model errors, they also conclude that the cause of this "stage 11 paradox" is to be attributed to an anomalous $CO_2$ rise.  They also observed that, CVAR being "largely linear", nonlinearities and thresholds do not play an important role in the major part of the latest 800,000 years, and reject the hypothesis that glacial-interglacial cycles could

occur without orbital forcing.

Statistical modelling indeed has much to offer for analysing and understanding glacial-interglacial cycles. It is a good approach for detecting an "anomaly" and then enquire about the causes of this anomaly.

I, however, see two problems with the present contribution.

First: reading is tedious, notations and tables not always clear, and the model description should be more self-contained. The second problem, perhaps more serious, is what I would consider a misuse of statistical reasoning and dynamical systems theory.

It should here be reminded that satisfactory model performance over most of the latest 800,000 years is not enough for rejecting alternatives. There is, in the present contribution, no statistically framed attempt at comparing models. Some alternative models actually do a fairly convincing job in reproducing the full sequence of the latest 800,000 years, including stage 11, and some of these models actually are limit cycle synchronised on the orbital forcing. Similarly, that one model can produce the full sequence of the last eight glacial-interglacial cycles without parameter change does not reject the hypothesis that a regime change actually occurred.

Furthermore, a linear dynamical system forced by harmonics (sines and cosines) can only output harmonics (you can show this by reasoning on the Fourier transform). The orbital forcing is a sum of harmonics. Hence, non-linearity is needed to transform this sum of sines and cosines into the characteristic saw-tooth-shaped, 100,000 year-long nature of glacial-interglacial cylces, which is most visible over the latest 400,000 years. So even though the authors qualify the CVAR model as "largely linear" (p. 9), it must nevertheless contain the nonlinearity necessary to reproduce these characteristics. Yet, the authors are silent on the consequences on this non-linearity, and in particular (for reproducing which features) when it is critical.

Inspecting figure 1, it seems that the CVAR is missing two important tests: the termination five (which the authors focus on), and the termination one leading to the current interglacial. Both occurred despite a relatively weak orbital forcing, and both actually justify the widely held assumption that the deglaciations involve non-linear dynamics, perhaps catastrophic dynamics (the idea that a "mature" glacial stage is unstable).

Line by line comments

Page 2, line 18: "reject the hypothesis that carbon dioxide or methane is exogenous to the climate system". It's not the purpose here to comment and criticize KJ2013, but I must, however, say that I find this statement puzzling, or at least misleading. No one seriously disputes that $CO_2$ is somehow generated and cycled within the earth, and in that sense it belongs to the climate system. Whether $CO_2$ dynamics, in a given model, is treated as endogenous or exogenous is not an ontological statement about the nature of the system. It is a working hypothesis that helps to addressing a specific question.

Page 3, line 6: "glacial cycles are driven by the same dynamics before and after the MBE". There may be a type confusion here. In what sense are dynamics "driving" something? In common language, a driver rather refers to an external agent (as in the sentence: "glacial-interglacial cycles are driven by orbital forcing")

Page 3, line 34: "six climate and four mechanisms": again, this seems to be a type confusion. How can a "variable" be a "mechanism"? Mechanism refers to a chain of causes, not to a variable (ditto page 4, lines 10-11).

Page 3, line 21: it is perfectly acceptable, for this kind of study, to adopt a common timescale, and EDC3 can indeed do the job, even though it has been subject to some revision, especially around stage 11 (see the AICC2012 time scale, Bazin et al. 2013, 10.5194/cp-9-1715-2013). However, synchronising $CO_2$ with sea-level records remains a challenging exercise, which could have implications for the interpretation of the results, especially when it comes to discussing the timing of model errors.

Page 5, line 5: the alpha matrix of coefficients seems to play a very important role in the dynamics of the model. It is presented as a matrix of relaxation coefficients, but later in the document it seems that these are not simple linear relaxations (page 9, line 21) since the authors mention that it is a source of non-linearity. Again, the authors should be more explicit about the construction of this matrix, and about the implications of the possible non-linearities.

Page 5, line 31: "values from 792 kyr BP through 392 kyr BP constitute the out of sample period". This seems to be the only place where the out-of-sample period is clearly defined. If I understood correctly, the whole statistical analyses supporting the present contribution is based on one out-of-sample period, and one in-sample one. What about swapping the in-sample and out-of-sample? Are conclusions unchanged?

Page 6, line 29: The $Y_{ji}$ (an index i) are not defined.

Page 9, line 9: "together, these results suggest that the test results reveal information about the statistical ordering of simulation errors" : I'm afraid that I could not make any sense of that sentence.

Page 11, lines 28 to 36: the mechanism discussion in fact mainly contains (with a few exceptions) references about the sequence of Heinrich events and Yourger Dryas, and not so much about the initiation of the deglaciation. Many of these articles are not related at all to stage 11.

Page 12, line 34: "together, these results suggest that terminations in general, and termination five in particular, are driven by changes in atmospheric carbon dioxide". First, the authors must clarify what they mean by "driven". The overall stance of the article is that orbital forcing is driving all "endogeneous" variables of the climate system, including carbon dioxide. But we can understand that the authors mean that the CO2 rise, whatever its cause, is a crucial element of the causal chain that leads to the deglaciation, and that "something" caused its rise, which is not ice melt or sea-level rise (this is the meaning which I could give to the sentence at the end of page 12 : "

they contradict the notion that changes in carbon dioxide are a positive feedback loop in Earth system as opposed to a cause of glacial terminations").

Now, that $CO_2$ is indeed involved in the dynamics of the deglaciation is largely accepted by the experts of ice-age dynamics. The question is which roles it plays in the acceleration of deglacial dynamics, compared to the mechanisms of glacial instability (isostasy, buttressing effects, accelerated ice flows) or yet other phenomena (dust accumulation for example). This is a long ongoing debate, which is being investigated by considerations about the physics of ice sheets in ocean circulation, and by careful inspection of the climate records. Statistical analysis as the one presented here is part of the investigation, but it requires more attention to uncertainties associated to the dating and interpretation of palaeoclimate records.

Furthermore, an early rise in $CO_2$ does not mean that it causes the deglaciation. Rises in $CO_2$ have been observed throughout the latest glacial interglacial cycle (associated with the so-called Antarctic warming events), and did not yield deglaciations. Whether a 10 ppm $CO_2$ increase has to be considered as "the" nudge which triggered runaway deglacial dynamics (e.g. whether it is a "proximal" cause, see Wolff et al. 2009, dx.doi.org/10.1038/ngeo442) is perhaps an interesting question, but it does not make it the explanation of the deglaciation.

Figure 4 : a legend within the figure would be helpful

Table 2 is very hard to read. What the "distribution among Marine isotope status" means is not obvious at all. What are the units, which reading should one make of these numbers and what are the implications? What is the meaning of persisting errors?

Title: Paleo-Climate is not standard spelling.

---

## Author Comment (AC5) · 23 Jul 2020

[11pt, a4paper]article

graphicx

**Response to Reviewer 3**
Robert K. Kaufmann & Felix Pretis

We thank the reviewer for their detailed review of the manuscript. Our responses are are described point-by-point below.

[Figure]

*This contribution is based on a statistical model called CVAR, standing for "cointe-gration vector autoregression". The model is calibrated on the latest four glacial-interglacial cycles, and then shown to fully reproduce the sequence of the latest eight glacial-interglacial cycles, except for the deglaciation leading to the stage 11. The au-thors conclude that this deglaciation, associated with the mid-Brunhes event, is to be considered as an anomalous event rather than marking a regime transition. By in-specting model errors, they also conclude that the cause of this "stage 11 paradox" is to be attributed to an anomalous $CO_2$ rise. They also observed that, CVAR being "largely linear", nonlinearities and thresholds do not play an important role in the major part of the latest 800,000 years, and reject the hypothesis that glacial-interglacial cy-cles could occur without orbital forcing. Statistical modelling indeed has much to offer for analysing and understanding glacial-interglacial cycles. It is a good approach for detecting an "anomaly" and then enquire about the causes of this anomaly. I, however, see two problems with the present contribution. First: reading is tedious, notations and tables not always clear, and the model description should be more self-contained. The second problem, perhaps more serious, is what I would consider a misuse of statistical reasoning and dynamical systems theory.*

We will edit the manuscript to make it more readable and self-contained.

*It should here be reminded that satisfactory model performance over most of the latest 800,000 years is not enough for rejecting alternatives. There is, in the present contri-bution, no statistically framed attempt at comparing models. Some alternative models actually do a fairly convincing job in reproducing the full sequence of the latest 800,000 years, including stage 11, and some of these models actually are limit cycle synchro-nised on the orbital forcing. Similarly, that one model can produce the full sequence of the last eight glacial-interglacial cycles without parameter change does not reject the hypothesis that a regime change actually occurred.*

As far as we know, our manuscript is the first to assess the accuracy of the simulation, both in- and out-of-sample using these statistical rigorous methods. As such, it goes beyond conclusions about models '*doing a fairly convincing job.*' The reviewer is correct; we do not explicitly compare results among models. Instead, conclusions about nonlinearities and thresholds are based on Occam's razor. Conditioning on only orbital geometry, our model is able to account for glacial cycles (both in- and out-of-sample) without nonlinearities and threshold effects. Occam's razor implies that if nonlinearities and/or threshold effects were critical, a linear model would be unable to simulate glacial cycles. As we state in the abstract (line 13-15), our results "suggests that nonlinearities and/or threshold effects do not play a critical role in glacial cycles."

Yes, "*that one model can produce the full sequence of the last eight glacial-interglacial cycles without parameter change does not reject the hypothesis that a regime change actually occurred*" but Occam's razor suggests that any regime change is not important. An important change would certainly limit the model's ability to accurately "*produce the full sequence of the last eight glacial-interglacial cycles.*"

*Furthermore, a linear dynamical system forced by harmonics (sines and cosines) can only output harmonics (you can show this by reasoning on the Fourier transform). The orbital forcing is a sum of harmonics. Hence, non-linearity is needed to transform this sum of sines and cosines into the characteristic saw-tooth-shaped, 100,000 year-long nature of glacial-interglacial cylces, which is most visible over the latest 400,000 years. So even though the authors qualify the CVAR model as "largely linear" (p. 9), it must nevertheless contain the nonlinearity necessary to reproduce these characteristics. Yet, the authors are silent on the consequences on this non-linearity, and in particular(for reproducing which features) when it is critical.*

Our linear model is able to "*transform this sum of sines and cosines into the characteristic saw-tooth-shaped, 100,000 year-long nature of glacial-interglacial cylces*" So,
non-linearity is not necessarily needed to transform this sum of sines and cosines into the characteristic saw-tooth-shaped. There is no 'hidden' nonlinear aspect of the model, however, we apologize; we should have been clearer regarding the linear nature of the CVAR model. The CVAR model is linear in parameters. That is, the first cointegrating relation in Table 2 of Kaufmann and Juselius (2013) and Supplementary Table S.1 is a linear relation between Temp and $CO_2$ and the third cointegrating relation is a linear relation among Ice, $CO_2$, and eccentricity. These linear long-run cointegrating relations, which lie at the heart of the CVAR, is what we refer to as a 'linear model'. We will clarify this in our revised manuscript (see also the response to Reviewer #2).

The apparent non-linear dynamics can stem from two sources. First, the individual variables (specifically, the first difference of each variable) such as temperature adjust towards disequilibrium as a linear function of disequilibrium in the level of the variables in the previous time period. This creates a seeming non-linear change in the level. However, the model is linear in both first differences and levels.

The reviewer's claim about a hidden nonlinear component can be addressed with the clever experiment suggested by Reviewer #2. He/she also was skeptical of a linear model's ability to simulate the climate record. To test this skepticism, he/she suggested that we use our CVAR model to estimate a model among endogenous variables generated by a van-der-Pol oscillator that is driven by an exogenous sine-wave function. If the CVAR model embodies nonlinear relations, anonymous reviewer #2 postulates that the CVAR model would be able to simulate the nonlinear data generating process. But as indicated in our response to anonymous Reviewer #2 (Figures 1-3), the CVAR model is not able to simulate the nonlinear data generating process in a statistically significant fashion. Instead, the results are consistent with the alternative hypothesis proposed by anonymous reviewer #2, that nonlinearities do not play a critical role in glacial cycles, because the CVAR model is able to account for glacial cycles in a statistically meaningful fashion both in- and -out-of-sample.

*Inspecting figure 1, it seems that the CVAR is missing two important tests: the termination five (which the authors focus on), and the termination one leading to the current interglacial. Both occurred despite a relatively weak orbital forcing, and both actually justify the widely held assumption that the deglaciations involve non-linear dynamics, perhaps catastrophic dynamics (the idea that a "mature" glacial stage is unstable).*

The manuscript does not ignore "*the termination one leading to the current interglacial.*". The CVAR model's inability to accurately simulate the last interglacial is described in the last two paragraphs of the Conclusion, which links both difficulties to atmospheric $CO_2$.

As indicated in Figure 1a, the statistical methodology indicates that the CVAR model does not simulate Ice accurately during the current interglacial. But this does not "*justify the widely held assumption that the deglaciations involve non-linear dynamics, perhaps catastrophic dynamics (the idea that a "mature" glacial stage is unstable).*" As described on page 13, the literature contains another testable hypothesis "Holocene warming is amplified by anthropogenic emissions of carbon dioxide and methane (Ruddiman 2003; 2005; 2007)." And this effect can be tested by our model. As stated on page 13, "These CVAR simulations also will be used to assess the early Anthropogenic hypothesis by evaluating the degree to which anthropogenic emissions of carbon dioxide and methane can account for outliers and persisting errors in Ice and other climate variables during the Holocene."

*Page 2, line 18: "reject the hypothesis that carbon dioxide or methane is exogenous to the climate system". It's not the purpose here to comment and criticize KJ2013, but I must, however, say that I find this statement puzzling, or at least misleading. No one seriously disputes that $CO_2$ is somehow generated and cycled within the earth, and in that sense it belongs to the climate system. Whether $CO_2$ dynamics, in a given model, is treated as endogenous or exogenous is not an ontological statement about*

*the nature of the system. It is a working hypothesis that helps to addressing a specific question.*

As the reviewer states "*It is a working hypothesis*" and that working hypothesis is rejected by our model, which is able to simulate $CO_2$ endogenously in a statistically significant fashion. We emphasize that the CVAR simulates $CO_2$ endogenously because the working hypothesis that $CO_2$ is exogenous undermines the results of many previous models. For models that treat $CO_2$ as an exogenous variable, analysts cannot separate the explanatory power of the model from the explanatory power of carbon dioxide. Observed values of atmospheric $CO_2$ can account for much of the variation n in the climate system, which is simply 'apportioned out' to the other components of the climate system by models conditioned on $CO_2$.

*Page 3, line 6: "glacial cycles are driven by the same dynamics before and after the MBE". There may be a type confusion here. In what sense are dynamics "driving" something? In common language, a driver rather refers to an external agent (as in the sentence: "glacial-interglacial cycles are driven by orbital forcing").*

Yes, the wording is imprecise. We will revise the sentence to read glacial cycles are driven by the same short- and long-run relations between orbital geometry and the climate system and among components of the climate system before and after the MBE.

*Page 3, line 34: "six climate and four mechanisms": again, this seems to be a type confusion. How can a "variable" be a "mechanism"? Mechanism refers to a chain of causes, not to a variable (ditto page 4, lines 10-11).*

Yes, the wording is imprecise. Some of the variables in the model are important indicators of the climate system per se, such as sea surface temperature and the extent of land ice. But other variables are important because they proxy aspects of the climate system. For example, the model includes Na because it proxies the extent of sea ice. We will modify the text accordingly.

*Page 3, line 21: it is perfectly acceptable, for this kind of study, to adopt a com-mon timescale, and EDC3 can indeed do the job, even though it has been subject to some revision, especially around stage 11 (see the AICC2012 time scale, Bazin et al.2013, 10.5194/cp-9-1715-2013). However, synchronising $CO_2$ with sea-level records remains a challenging exercise, which could have implications for the interpretation of the results, especially when it comes to discussing the timing of model errors.*

This hypothesis is investigated by Kaufmann and Juselius (2016), who find that "This result is unaffected by astronomically tuned data; repeating the analysis with a time series for land ice that is not tuned (Huybers, personal communication) generates similar results regarding the nature of the tenth cointegration relation (results available from the authors)." We will include this result when we revise the manuscript.

*Page 5, line 5: the alpha matrix of coefficients seems to play a very important role in the dynamics of the model. It is presented as a matrix of relaxation coefficients, but later in the document it seems that these are not simple linear relaxations (page 9, line 21) since the authors mention that it is a source of non-linearity. Again, the authors should be more explicit about the construction of this matrix, and about the implications of the possible non-linearities.*

As part of the CVAR model, the matrix of error correction coefficients (equation 3 on

page 5) is estimated statistically. As indicated by equations (2) and (3), individual variables (specifically, the first difference of each variable) such as temperature adjust towards disequilibrium as a linear function of disequilibrium in the level of the variables in the previous time period. This creates a seeming non-linear change in the level. However, the model is linear in parameters in both first differences and levels (see also our response on non-linearity above).

*Page 5, line 31: "values from 792 kyr BP through 392 kyr BP constitute the out of sample period". This seems to be the only place where the out-of-sample period is clearly defined. If I understood correctly, the whole statistical analyses supporting the present contribution is based on one out-of-sample period, and one in-sample one. What about swapping the in-sample and out-of-sample? Are conclusions unchanged?*

This is an interesting idea, but it is not possible to swap the in- and out-of-sample periods because of limits on the availability of data. As indicated in Table 1, 466 kyr BP is the first observation for proxy data for sea level from Siddal et al., (2003). As such, it would not be possible to estimate the CVAR for the 792 kyr BP through 392 kyr BP out-of-sample period. This highlights the strength of the out-of-sample period. The model is not initialized on observations for sea level (and $CO_2$, CH4, Fe, Na, SO4, Ca, and Level). Rather, the model is initialized with values that correspond to their sample mean and spun-up from 800-792 Kyr BP. Given this procedure, the model's accuracy starting in 792 Kyr BP appears remarkable to us. Only Na and SO4 fail tests of statistical accuracy at the start of the out-of-sample period.

*Page 6, line 29: The $Y_{ji}$ (an index i) are not defined.*

We will correct this in the revised manuscript.

*Page 9, line 9: "together, these results suggest that the test results reveal information about the statistical ordering of simulation errors" : I'm afraid that I could not make any sense of that sentence.*

To address the reviewer's comment, we will clarify this section.

*Page 11, lines 28 to 36: the mechanism discussion in fact mainly contains (with a few exceptions) references about the sequence of Heinrich events and Yourger Dryas, and not so much about the initiation of the deglaciation. Many of these articles are not related at all to stage 11.*

We will re-read these papers and eliminate those that are not related to stage11.

*Page 12, line 34: "together, these results suggest that terminations in general, and termination five in particular, are driven by changes in atmospheric carbon dioxide". First, the authors must clarify what they mean by "driven". The overall stance of the article is that orbital forcing is driving all "endogeneous" variables of the climate system, including carbon dioxide. But we can understand that the authors mean that the $CO_2$ rise, whatever its cause, is a crucial element of the causal chain that leads to the deglaciation, and that "something" caused its rise, which is not ice melt or sea-level rise (this is the meaning which I could give to the sentence at the end of page 12 : they contradict the notion that changes in carbon dioxide are a positive feedback loop in Earth system as opposed to a cause of glacial terminations").*

This question may be caused by our lack of clarity regarding "the statistical ordering of simulation errors" and "driven by changes in atmospheric carbon dioxide." We use statistical techniques to identify the timing of model errors. By doing so, we try to

identify the variable(s) that the model fails to simulate at a particular time and trace if and how this error propagates through the system. This strategy is explained on lines 9-14 on page 12:

> "These competing hypothesis for the terminations in general and stage 11 in particular can be tested by the statistical ordering of the model errors. If changes in sea surface temperature initiate Termination V, the model's inability to simulate termination V will 'start' with its inability to simulate SST. This inability will be indicated by simulation errors for SST that precede and have information about the simulation errors for other variables. Specifically, simulation errors for other variables, such as $CO_2$, will not have prior information about the errors for SST and these errors will have prior information about the errors for the other variables, such as $CO_2$."

Beyond tracing these errors, we also condition the model on SST, Na, Ca, or $CO_2$, which proxy competing hypotheses for the Stage 11 paradox. As indicated in Figure 4, observed values for atmospheric $CO_2$ have considerably more explanatory power for Ice during stage 11 that the other variables. This result suggests that "that terminations in general, and termination V in particular, are driven by changes in atmospheric carbon dioxide." This result goes back to our response about models treating $CO_2$ as an exogenous variable. Figure 4 suggests that if our model is conditioned on $CO_2$ (rather than simulating it endogenously), the model would no longer perform poorly during stage 11.

*Now, that $CO_2$ is indeed involved in the dynamics of the deglaciation is largely accepted by the experts of ice-age dynamics. The question is which roles it plays in the acceleration of deglacial dynamics, compared to the mechanisms of glacial instability (isostasy, buttressing effects, accelerated ice flows) or yet other phenomena (dust ac-*

*cumulation for example). This is a long ongoing debate, which is being investigated by considerations about the physics of ice sheets in ocean circulation, and by careful inspection of the climate records. Statistical analysis as the one presented here is part of the investigation, but it requires more attention to uncertainties associated to the dating and interpretation of palaeoclimate records.*

As described above, results are not changed when the model is estimated using the time series for Ice that is not orbitally tuned. Also, our model would not simulate interglacial periods accurately (other than stage 11) if isostasy, buttressing effects, accelerated ice flows played a critical role because these mechanisms are not explicitly represented by the CVAR model. It may also be the case that the time-steps used in our analysis (1k years) mask some of the perhaps 'faster' glacial events such as accelerated ice flows.

*Furthermore, an early rise in $CO_2$ does not mean that it causes the deglaciation. Rises in $CO_2$ have been observed throughout the latest glacial interglacial cycle (associated with the so-called Antarctic warming events), and did not yield deglaciations. Whether a 10 ppm $CO_2$ increase has to be considered as "the" nudge which triggered run-away deglacial dynamics (e.g. whether it is a "proximal" cause, see Wolff et al. 2009) is perhaps an interesting question, but it does not make it the explanation of the deglaciation.*

As described above, we are not examining the rise or fall in any variable, such as an early rise in $CO_2$. We are examining the statistical ordering among model errors. To summarize, the CVAR model fails to simulate Stage 11 in a statistically meaningful fashion. For what variable(s) does this general failure first appear. Tracing the error back to a variable would suggest that the model fails to account for an important mechanism, and this failure spreads to the other variables, and ultimately causes the stage 11 failure in the CVAR. In this case, the source of model failure during stage 11 is

the failure to simulate $CO_2$, as indicated by this error proceeding the errors for other variables.

*Figure 4 : a legend within the figure would be helpful*

We are happy to add a legend in the revised version.

*Table 2 is very hard to read. What the "distribution among Marine isotope status" means is not obvious at all. What are the units, which reading should one make of these numbers and what are the implications? What is the meaning of persisting errors?*

We will clarify this in the revised manuscript. This table shows the results of tests that outliers and persisting errors are distributed randomly between the in- and out-of-sample periods and among the 19 marine isotope stages. The procedure that generated these results are described in Section 2.4 Identifying Periods of Simulation Failure on page 6. Outliers and persisting errors are defined on pages 5-6 in Section 2.3 Statistical Measures of Model Performance "Outliers refer to a change in the simulated value of variable x relative to the observed value for a single time step, while persisting errors are statistically significant differences that persist for two or more consecutive time-steps."

*Title: Paleo-Climate is not standard spelling.*

We will revise this.